# Inter-comparison of daily precipitation products for large-scale hydro-climatic applications over Canada

Jefferson S. Wong[1*], Saman Razavi[1], Barrie R. Bonsal[2], Howard S. Wheater[1], & Zilefac E. Asong[1]

[1] *Global Institute for Water Security and School of Environment and Sustainability, University of Saskatchewan, 11 Innovation Blvd, Saskatoon, SK, Canada S7N 3H5*

[2] *Environment and Climate Change Canada, 11 Innovation Blvd, Saskatoon, SK, Canada S7N 3H5*

[*] Corresponding author:

Email: jefferson.wong@usask.ca

Phone: +1 306 966 7816

Abstract
A number of global and regional gridded climate products based on multiple data sources are
available that can potentially provide reliable estimates of precipitation for climate and
hydrological studies. However, research into the consistency of these products for various regions
has been limited and in many cases non-existent. This study inter-compares several gridded
precipitation products and quantifies the spatial and temporal variability of the errors (relative to
station observations) over 15 terrestrial ecozones in Canada for different seasons over the period
1979 to 2012 at a 0.5° and daily spatiotemporal resolution. These datasets were assessed in their
ability to represent the daily variability of precipitation amounts by four performance measures:
percentage of bias, root-mean-square-error, correlation coefficient, and standard deviation ratio.
Results showed that most of the datasets were relatively skillful in central Canada. However, they
tended to overestimate precipitation amounts in the west and underestimate in the north and east,
with the underestimation being particularly dominant in northern Canada (above 60° N). The
global product by WATCH Forcing Data ERA-Interim (WFDEI) augmented by Global
Precipitation Climatology Centre (GPCC) data (WFDEI [GPCC]) performed best with respect to
different metrics. The Canadian Precipitation Analysis (CaPA) product performed comparably
with WFDEI [GPCC], however it only provides data starting in 2002. All the datasets performed
best in summer, followed by autumn, spring, and winter in order of decreasing quality. Findings
from this study can provide guidance to potential users regarding the performance of different
precipitation products for a range of geographical regions and time periods.
**Keywords:** precipitation; evaluation and comparison; datasets; ecozones; hydro-climatology;
Canada

## 1. **Introduction**

The availability of accurate data, especially precipitation is essential for understanding the climate system and hydrological processes since it is a vital element of the water and energy cycles and a key forcing variable for driving hydrological models. Reliable precipitation measurements provide valuable information for meteorologists, climatologists, hydrologists, and other decision makers in many applications, including climate and/or land-use change studies (e.g. Cuo et al., 2011;Huisman et al., 2009;Dore, 2005), agricultural and environmental research (e.g. Zhang et al., 2012;Hively et al., 2006), natural hazards (e.g. Taubenbock et al., 2011;Kay et al., 2009;Blenkinsop and Fowler, 2007), and hydrological and water resources planning (e.g. Middelkoop et al., 2001;Hong et al., 2010). With respect to land-surface hydrology, the increasing sophistication of distributed hydrological modeling has urged the requirement of better and more reliable gridded precipitation estimates at a minimum, daily temporal resolution. Before incorporating precipitation measurements, quantifying their uncertainty becomes an essential prerequisite for hydrological applications and is increasingly critical for potential users who are left without guidance and/or confidence in the myriad of products for their specific hydrological problems over different geographical regions. This study attempts to address this issue by comparing and examining the error characteristics of different types of gridded precipitation products and assessing how these products perform geographically and temporally over Canada.

*Precipitation measurements and their limitations*

With technological and scientific advancements over the past three decades, tremendous progress has been made in the various methods of precipitation measurement, each one with its own strengths and limitations. Conventional measurements through the use of rain gauges continue to play an important role in precipitation observations, as they are the only source that provide the direct physical readings with relatively accurate measurements at specific points. However, such measurements are subject to various errors arising from wind effects (Nešpor et al., 2000;Ciach, 2003), evaporation (Strangeways, 2004;Mekis and Hogg, 1999), undercatch (Yang et al., 1998;Adam and Lettenmaier, 2003;Mekis and Hogg, 1999), and instrumental problems including basic mechanical and electrical failure. Moreover, since many applications such as distributed hydrological and hydraulic models require areal precipitation estimates, rain-gauge measurements are often spatially interpolated. Interpolation, however, may not capture the true spatial variability

of precipitation fields due to sparse gauge networks, particularly in complex terrains like
mountainous regions or remote high latitudes. Radars, as alternative ground-based measurements
can estimate precipitation over a relatively large area (radius of 200 to 300 km), but are also prone
to inaccuracies as a result of beam spreading, curvature of the earth, and terrain blocking (Dinku
et al., 2002;Young et al., 1999), and errors in the rain rate-reflectivity relationship, range effects,
and clutter (Jameson and Kostinski, 2002;Villarini and Krajewski, 2010). Development of
satellite-based precipitation estimates has provided coverage over vast gauged/ungauged regions
with continuous observations regardless of time of day, terrain, and weather condition of the
ground (Gebregiorgis and Hossain, 2015). The recently launched Global Precipitation
Measurement (GPM) Core Observatory has further opened up new opportunities for observing
worldwide precipitation from space (Hou et al., 2014). However, satellite-based estimates also
contain inaccuracies resulting primarily from temporal sampling errors due to infrequent satellite
visits to a particular location, instrumental errors due to calibration and measurement noise, and
algorithm errors related to approximations in cloud physics (Nijssen and Lettenmaier,
2004;Gebremichael et al., 2005). In particular, the passive microwave overpasses were shown to
be unreliable over regions with snow cover and complex terrain such as the Tibetan Plateau (Yong
et al., 2015).
Recognizing the limitations in the various precipitation observation methods, a number of attempts
to combine information from multiple sources have been undertaken (Xie and Arkin,
1996;Maggioni et al., 2014;Shen et al., 2010). Numerous approaches were developed to produce
high-resolution estimates through combining infrared and microwave data (e.g. Huffman et al.,
2007;Turk et al., 2010), merging multi-satellite products with gauge observations (e.g. Huffman
et al., 1997;Huffman et al., 2010;Adler et al., 2003;Xie and Arkin, 1997;Wang and Lin, 2015), and
implementing different precipitation retrieval techniques (e.g. Joyce et al., 2004;Hsu et al., 2010).
Reanalysis data provide an alternative source of precipitation estimates that mitigate the sparse
distribution of observations by assimilating all available data (rain-gauge stations, aircraft, satellite,
etc.) into a background forecast physical model. However, they are only an estimate of the real
state of the atmosphere which do not necessarily match the observations (Bukovsky and Karoly,
2007;West et al., 2007). Inaccuracies in reanalysis precipitation might also arise from the complex
interactions between the model and observations that depend on the specific analysis-forecast
systems and the choice of physical parameterizations, especially in regions with missing
observations (Betts et al., 2006). Numerical climate models including Atmosphere-Ocean General
Circulation Models (AOGCMs) and Regional Climate Models (RCMs) offer another potential
source of precipitation estimates, as well as future precipitation simulations. AOGCMs remain
relatively coarse in resolution (approximately 100 to 250 km) and are not able to resolve important
sub-grid scale features such as topography, land cover, and clouds (Grotch and Maccracken, 1991),
resulting in the requirement of downscaling to provide fine resolution climate parameters for
hydrological analyses. In general, precipitation estimates from climate models often produce
systematic bias due to imperfect model-conceptualization, discretization and spatial averaging
within grid cells (Teutschbein and Seibert, 2010;Xu et al., 2005).
*Scope and Objectives*
Numerous previous evaluation efforts among the precipitation products have been limited into
three groups of inter-comparison of (1) satellite-derived products (e.g. Adler et al., 2001;Xie and
Arkin, 1995;Turk et al., 2008); (2) reanalysis data (e.g. Janowiak et al., 1998;Bosilovich et al.,
2008;Betts et al., 2006;Bukovsky and Karoly, 2007); and (3) climate model simulations (e.g.
Covey et al., 2003;Christensen et al., 2007;Mearns et al., 2006;2012). Despite the aforementioned
efforts, few studies have conducted a detailed inter-comparison among different types of
precipitation products. Gottschalck et al. (2005) compared seasonal total precipitation of several
satellite-derived, rain-gauge-based, and model-simulated datasets over contiguous United States
and showed the spatial root mean square error of seasonal total precipitation and mean correlation
of daily precipitation between each product and the impacts of these errors on land surface
modelling. Additionally, Ebert et al. (2007) examined 12 satellite-derived precipitation products
and four numerical weather prediction models over the United States, Australia, and northwestern
Europe and found that satellite-derived estimates performed best in summer and model-induced
ones were best in winter. However, a number of questions regarding the reliability of the
precipitation products remained in doubt, including: to what extent do the users have the
knowledge about the error information associated with all these different types of precipitation
products; how do the error distribution of precipitation products vary by location and season; and
which product(s) should the users have more confidence for their regions of interest. Answering
these questions is therefore a crucial first step in quantifying the spatial and temporal variability
of the precipitation products so as to better understand their reliability as forcing inputs in
hydrological modelling and other related studies.
Given the emergence of various products derived from different methods and sources (Tapiador
et al., 2012), accuracy comparison studies of precipitation products have been reported over
several regions; examples include the globe (e.g. Gebregiorgis and Hossain, 2015;Adler et al.,
2001;Tian and Peters-Lidard, 2010), Europe (e.g. Frei et al., 2006;Chen et al., 2006;Kidd et al.,
2012), Africa (e.g. Dinku et al., 2008;Asadullah et al., 2008), North America (e.g. Tian et al.,
2009;West et al., 2007), South America (e.g. Vila et al., 2009), and China (e.g. Shen et al.,
2010;Wetterhall et al., 2006). However, less attention has been paid to high-latitude regions such
as Canada where a considerable proportion of precipitation is in the form of snow (Behrangi et al.,
2016). In many regions of Canada, precipitation-gauge stations are sparsely distributed and the
information required for hydrological modelling may not be available at the site of interest. This
is especially true in northern areas (north of 60° N) and over mountainous regions where
precipitation-gauge stations are usually 500 to 700 km apart or at low elevations (Wang and Lin,
2015). Meanwhile, the decline and closure of manual observing precipitation-gauge stations
further reduced the spatial coverage and availability of long-term precipitation measurements
(Metcalfe et al., 1997;Mekis and Hogg, 1999;Rapaic et al., 2015). Of additional concern, the
observations for solid precipitation (snow, snow pellets, ice pellets, and ice crystals) and
precipitation phase (liquid or solid) changes make accurate measurement of precipitation more
difficult and challenging, and the measurement errors have been found to range from 20 to 50 %
for automated systems (Rasmussen et al., 2012). The Meteorological Service of Canada has
implemented a network of 31 radars (radar coverage at full range of 256 km) along southern
Canada (see Fortin et al. (2015b) Fig. 1 for spatial distribution). This Canadian radar network has
been employed as an additional source of observations in generating a gridded product for Canada
(see Sect. 3.2.2 for details). Yet, the shortcomings of using the radar data are twofold: (1) many
areas of the country (north of 60° N) are not covered by this network; and (2) the implementation
of the network began in 1997 and thus did not have sufficient lengths of data for any long-term
hydro-climatic studies. The availability, coverage, and quality of precipitation-gauge
measurements are thus obstacles to effective hydrological modelling and water management in
Canada. However, the availability of several global and regional gridded precipitation products
which provide complete coverage of the whole country at applicable time and spatial scales may
provide a viable alternative for regional- to national-scale hydrological applications in Canada.
Given the aforementioned, this study aims to (1) inter-compare various daily gridded precipitation
products against the best available precipitation-gauge observations; and (2) characterize the error
distributions of different types of precipitation products over time and different geographical
regions in Canada. Such inter-comparison will in turn help assess the performance of the
precipitation products over specific climatic/hydrological regions.
The rest of this paper is organized as follows: a brief description of the study area and precipitation
data is provided in Sect. 2 and 3. The methodology for evaluating precipitation products against
the precipitation-gauge station observations is described in Sect. 4. Results and discussion are
provided in Sect. 5 and 6, respectively, with a summary and conclusion following in Sect. 7.
2.   **Study Area**
Canada, which covers a land area of 9.9 million km$^2$, extends from 42° N to 83° N latitude and
spans between 141° W to 52° W longitude. With substantial variations over its landmass, the
country can be divided into many regions according to aspects such as climate, topography,
vegetation, soil, geology, and land use. The National Ecological Framework for Canada classified
ecologically distinct areas with four hierarchical levels of generalization (15 ecozones, 53
ecoprovinces, 194 ecoregions, and 1021 ecodistricts from broadest to the smallest) (Ecological
Stratification Working Group, 1996;Marshall et al., 1999). Similarly, the Standard Drainage Area
Classification (SDAC) was developed to delineate hydrographic areas to cover all the land and
interior freshwater lakes of the country with three levels of classification (11 major drainage areas,
164 sub-drainage areas, and 974 sub-sub-drainage areas) (Brooks et al., 2002;Pearse et al., 1985).
The precipitation comparisons in this study incorporated both the ecological and hydrological
delineations. This involved classifying the Canadian landmass into 15 ecozones for the main study
(Fig. 1) and 14 major drainage areas (the Arctic Major Drainage Area was further divided into
Arctic and Mackenzie, whereas the St. Lawrence Major Drainage Area was further split into St.
Lawrence, Great Lakes, and Newfoundland). Results are based on the ecozone classification, while
those based on drainage areas are reported in the supplementary material.
3.   **Precipitation Data**

## 3.1. **Precipitation-gauge station observations**

In Canada, climate data collection is coordinated by the Federal government, which is made available by the National Climate Data Archive of Environment and Climate Change Canada (NCDA). These data provide the basis for all available quality controlled climate observations. There are a total of 1499 precipitation-gauge stations (as of 2012) across Canada. However, given the frequent addition and subtraction of climate stations, these numbers have greatly varied through time with peak reporting in the 1970s followed by a general decline to the present (see Hutchinson et al. (2009) Figs. 1 and 2 for details). Furthermore, the existing precipitation observations are often subject to various errors, with gauge undercatch being of significant concern (Mekis and Hogg, 1999). To account for various measurement issues, Mekis and Hogg (1999) first produced the Adjusted and Homogenized Canadian Climate Data (AHCCD) including adjusted daily rainfall and snowfall values and Mekis and Vincent (2011) then updated the data for a subset of 464 stations over Canada. The data extend back to 1895 for a few long-term stations and run through 2014. As a result of adjustments, total rainfall amounts were on the order of 5 to 10 % higher in southern Canada and more than 20 % in the Canadian Arctic when compared to the original observations. Adjustments to snowfall were even larger and varied throughout the country. These adjusted values are widely considered as better estimates of actual precipitation and therefore have been used in numerous analyses (e.g. Nalley et al., 2012;Shook and Pomeroy, 2012;Wan et al., 2013;Asong et al., 2015). Given the lack of an adjusted daily gridded precipitation product for Canada, the AHCCD station precipitation is considered to be the best available data for Canada and thus is used as the benchmark for all gridded precipitation product comparisons.

## 3.2. **Gridded precipitation products**

Seven precipitation datasets were chosen for assessment based on the following criteria: (1) a complete coverage of Canada; (2) minimum of daily temporal and 0.5° (~50 km) spatial resolutions; (3) sufficient length of data (>30 years) for long-term study including recent years up to 2012; and (4) representing a range of sources/methodologies (e.g. station based, remote sensing, model, blended products). Table 1 summarizes these datasets, including their full names and original spatial and temporal resolutions for the versions used. Note that other commonly used datasets including the monthly Canadian Gridded temperature and precipitation (CANGRD) (Zhang et al., 2000), the coarser resolution Japan Meteorological Agency 55-year Reanalysis

(JRA-55) (Onogi et al., 2007;Kobayashi et al., 2015), and the Modern-Era Retrospective Analysis
for Research and Applications (MERRA) (Rienecker et al., 2011) products were excluded as they
do not meet criteria (2) above.

### 3.2.1. Station-based product – ANUSPLIN

Hutchinson et al. (2009) used the Australian National University Spline (ANUSPLIN) model to
develop a dataset of daily precipitation, and daily minimum and maximum air temperature over
Canada at a spatial resolution of 300 arc-seconds (0.0833° or ~10 km) for the period of 1961 to
2003. All available NCDA stations (that ranged from 2000 to 3000 for any given year during this
period) were used an input to the gridding procedure. To retain maximum spatial coverage, the
smaller number of stations in AHCCD were not incorporated (i.e. only unadjusted archive values
were used). Interpolation procedures included incorporation of tri-variate thin-plate smoothing
splines using spatially continuous functions of latitude, longitude, and elevation. Hopkinson et al.
(2011) subsequently extended this original dataset to the period 1950 to 2011. The Canadian
ANUSPLIN has now further been updated to 2013 and has recently been used as the basis of
'observed' data for evaluating different climate datasets (e.g. Eum et al., 2012) and for assessing
the effects of different climate products in hydro-climatological applications (e.g. Eum et al.,
2014;Bonsal et al., 2013;Shrestha et al., 2012a).

### 3.2.2. Station-based multiple-source product – CaPA

In November 2003, the Canadian Precipitation Analysis (CaPA) was developed to produce a
dataset of 6-hourly precipitation accumulation over North America in real-time at a spatial
resolution of 15 km (from 2002 onwards) (Mahfouf et al., 2007). Data were generated using an
optimum interpolation technique (Daley, 1993), which required a specification of error statistics
between observations and a background field (e.g. Bhargava and Danard, 1994;Garand and
Grassotti, 1995). For Canada, the short-term precipitation forecasts from the Canadian
Meteorological Centre (CMC)'s regional Global Environmental Multiscale (GEM) model (Cote
et al., 1998a;1998b) were used as the background field with the rain-gauge measurements from
NCDA as the observations to generate an analysis error at every grid point. CaPA become
operational at the CMC in April 2011, with updates in the statistical interpolation method
(Lespinas et al., 2015) and increase of spatial resolution to 10 km. The assimilation of Quantitative
Precipitation Estimates from the Canadian Weather Radar Network is also used as an additional
source of observations (Fortin et al., 2015b). With its continuous improvement and different
configurations, CaPA has been employed in Canada for various environmental prediction
applications (e.g. Eum et al., 2014;Fortin et al., 2015a;Pietroniro et al., 2007;Carrera et al., 2015).
However, the study period of these applications only start in 2002.
3.2.3. **Reanalysis-based multiple-source products – Princeton, WFDEI, and NARR**
*Princeton*
The Terrestrial Hydrology Research Group at the Princeton University initially developed a dataset
of 3-hourly near-surface meteorology with global coverage at 1.0° spatial resolution (~120 km)
from 1948 to 2000 for driving land surface models and other terrestrial systems (Sheffield et al.,
2006). This dataset (called hereafter "Princeton") was constructed based on the National Centers
for Environmental Prediction-National Center for Atmospheric Research (NCEP-NCAR)
reanalysis (2.0° and 6-hourly) (Kalnay et al., 1996;Kistler et al., 2001), combined with a suite of
global observation-based data including the Climatic Research Unit (CRU) monthly climate
variables (New et al., 1999, 2000), the Global Precipitation Climatology Project (GPCP) daily
precipitation (Huffman et al., 2001), the Tropical Rainfall Measuring Mission (TRMM) 3-hourly
precipitation (Huffman et al., 2002), and the NASA Langley Research Center monthly surface
radiation budget (Gupta et al., 1999). With the inclusion of additional temperature and
precipitation data (e.g. Willmott et al., 2001), Princeton has been updated and is currently available
with two versions: 1) 1948 to 2008 at 1.0°, 0.5°, and 0.25° at 3-hourly, daily, and monthly time
steps and 2) 1901-2012 experimental version at 1.0° and 0.5° at 3-hourly, daily, and monthly time
steps (used in this study). Studies employing Princeton to examine different hydrological aspects
have been carried out over different parts of Canada. For instance, Kang et al. (2014) examined
the changing contribution of snow to runoff generation in the Fraser River Basin while Su et al.
(2013) investigated the relationships between spring snow and warm-season precipitation in
central Canada. In addition, Wang et al. (2013) and Wang et al. (2014) used this dataset to
characterize the spatial and seasonal variations of the surface water budget at Canada national scale.
*WFDEI*
To simulate the terrestrial water cycle using different land surface models and general hydrological
models, the European Union Water and Global Change (WATCH) Forcing Data (WFD) were
created to provide datasets of sub-daily (3- and 6-hourly) and daily meteorological data with global
coverage at 0.5° spatial resolution (~50 km) from 1901 to 2001 (Weedon et al., 2011). Similar to
Princeton, the WFD were derived from the 40-year European Centre for Medium-Range Weather
Forecasts (ECMWF) Re-Analysis (ERA-40) (1.0° and 3-hourly) (Uppala et al., 2005) and
combined with the CRU monthly variables and the Global Precipitation Climatology Centre
(GPCC) monthly data (Rudolf and Schneider, 2005;Schneider et al., 2008;Fuchs, 2009). The
WATCH Forcing Data methodology applied to ERA-Interim (WFDEI) dataset has further been
developed covering the period of 1979 to 2012 (Weedon et al., 2014). The WFDEI used the same
methodology as the WFD, but was based on the ERA-Interim (Dee et al., 2011) with higher spatial
resolution (0.7°). As for the WFD, the WFDEI had two sets of rainfall and snowfall data generated
by using either CRU or GPCC precipitation totals. Both sets of data were used in this study
(hereafter known as WFDEI [CRU] and WFDEI [GPCC], respectively). To date, specific studies
using the WFDEI related to Canada have been limited to the investigation of permafrost changes
in the Arctic regions (e.g. Chadburn et al., 2015;Park et al., 2015;Park et al., 2016).
*NARR*
With the aim of evaluating spatial and temporal water availability in the atmosphere, the North
American Regional Reanalysis (NARR) was developed to provide datasets of 3-hourly
meteorological data for the North America domain at a spatial resolution of 32 km (~0.3°) covering
the period of 1979 to 2003 as the retrospective system and is being continued in near real-time
(currently up to 2015) as the Regional Climate Data Assimilation System (R-CDAS) (Mesinger et
al., 2006). The components in generating NARR included the NCEP-DOE reanalysis (Kanamitsu
et al., 2002), the NCEP regional Eta Model (Mesinger et al., 1988;Black, 1988) and the Noah land-
surface model (Mitchell et al., 2004;Ek et al., 2003), and the use of numerous additional data
sources (see Mesinger et al., 2006 Table 2). For hydrological modelling in Canada, Choi et al.
(2009) found that NARR provided reliable climate inputs for northern Manitoba while Woo and
Thorne (2006) concluded that NARR had a cold bias resulting in later snowmelt peaks in subarctic
Canada. In addition, Eum et al. (2012) identified a structural break point in the NARR dataset
beginning in January 2004 over the Athabasca River basin due to the assimilation of station
observations over Canada being discontinued in 2003.
3.2.4. **GCM statistically downscaled products – PCIC**
The Pacific Climate Impacts Consortium (PCIC), which is a regional climate service centre at the
University of Victoria, British Columbia, Canada, has offered datasets of statistically downscaled
daily precipitation and daily minimum and maximum air temperature under three different
Representative Concentration Pathways (RCPs) scenarios (RCP 2.6, 4.5, and 8.5) (Meinshausen
et al., 2011) over Canada at a spatial resolution of 300 arc-seconds (0.833° or ~10 km) for the
historical and projected period of 1950 to 2100 (Pacific Climate Impacts Consortium; University
of Victoria, Jan 2014). These downscaled datasets were a composite of 12 GCM projections from
the Coupled Model Inter-comparison Project Phase 5 (CMIP5) (Taylor et al., 2012) and the
ANUSPLIN dataset. The historical 1950 to 2005 period of the ANUSPLIN was used for bias-
correction and downscaling of the GCMs. Two different methods were used to downscale to a
finer resolution (Werner and Cannon, 2016). These included Bias Correction Spatial
Disaggregation (BCSD) (Wood et al., 2004) following Maurer and Hidalgo (2008) and Bias
Correction Constructed Analogues (BCCA) with Quantile mapping reordering (BCCAQ), which
was a post-processed version of BCCA (Maurer et al., 2010). The ensemble of the PCIC dataset
has currently been used in studying the hydrological impacts of climate change on river basins
mainly in British Columbia (e.g. Shrestha et al., 2011;Shrestha et al., 2012b;Schnorbus et al., 2014)
and Alberta (e.g. Kienzle et al., 2012;Forbes et al., 2011) in Canada. In this study, only four GCMs
with two respective statistical downscaling methods were chosen for comparison (see Table 2 for
details). The choice of the four GCMs was to match those available in the NA-CORDEX dataset
(see next section for details).
3.2.5. **GCM-driven RCM dynamically downscaled products – NA-CORDEX**
Sponsored by the World Climate Research Programme (WCRP), the COordinated Regional
climate Downscaling EXperiment (CORDEX) over North America domain (NA-CORDEX)
provides dynamically downscaled datasets of 3-hourly or daily meteorological data over most of
North America (below 80° N) at spatial resolutions of 0.22° and 0.44° (~25 and ~50 km) under
RCP 4.5 and 8.5 for the historical (1950 – 2005) and future (2006 – 2100) period (Giorgi et al.,
2009). Drawing from the strengths of the North American Regional Climate Change Assessment
Program (NARCCAP) (Mearns et al., 2012), a matrix of six GCMs from the CMIP5 driving six
different RCMs was selected to compare and characterize the uncertainties of RCMs and thus
provided climate scenarios for further impact and adaption studies. Current studies using NA-
CORDEX datasets were mainly focused on evaluating the model performance of different GCM-
driven RCM simulations over North America (e.g. Lucas-Picher et al., 2013;Martynov et al.,
2013;Separovic et al., 2013). In this study, two GCMs and three RCMs were chosen for
comparison due to the availability of the NA-CORDEX dataset (see Table 3 for details).
4.  **Methodology**
4.1. **Pre-processing**
Due to the different spatial and temporal resolutions of the various precipitation products, the first
step was to re-grid each onto a common 0.5° x 0.5° resolution to match the lowest-resolution
dataset. It was acknowledged that re-gridding products onto a common spatial resolution might
introduce more errors or uncertainties and the number of interpolation steps should be minimized.
However, the main focus of this study was to inter-compare various gridded precipitation products
using precipitation-gauge station data as a reference/benchmark but not to assess the individual
accuracy of each product against the reference dataset. Therefore, upscaling to a common
resolution provided a direct and more consistent inter-comparison. Such methodology was
consistent with similar studies in the literature (e.g. Janowiak et al., 1998;Rauscher et al.,
2010;Kimoto et al., 2005). All data were accumulated to daily time scale for comparison. Two
common time spans were selected since CaPA covered a shorter timeframe compared to the rest
of the products: (1) long-term comparison from January 1979 to December 2012 with the exclusion
of CaPA (from January 1979 to December 2005 for PCIC and NA-CORDEX as the historical
period of the datasets ends in 2005); and (2) short-term comparison from January 2002 to
December 2012 when CaPA data are available. Daily values were summed over the four standard
seasons (spring: March to May – MAM, summer: June to August – JJA, autumn: September to
November – SON, and winter: December to February – DJF) to inter-compare the precipitation
products at a seasonal scale.
To identify the most consistent gridded dataset corresponding to different seasons and regions,
comparisons of each dataset with direct precipitation-gauge station data from the aforementioned
AHCCD were carried out. For the period of 1979 to 2012, only 169 of the original 464 stations
across Canada were available. This drastic drop was due to 271 stations ending before or after
early 2000s and 23 not having a complete year of 2012. Subsequently, any of the 169 stations
where the percentage of missing values exceeded 10 % during the study period were also
eliminated. This resulted in 145 and 137 stations across Canada for long-term and short-term
comparison respectively (see Fig. 1 for locations). Note that most of the stations are located in
southern Canada with only 15 stations above 60° N.
Gridded-based precipitation estimates at the coordinates of the precipitation-gauge stations were
then extracted by employing an inverse-distance-square weighting method (Cressman, 1959),
which has been used to interpolate climate data for simple and efficient applications (Eum et al.,
2014;Shen et al., 2001). This method assumes that an interpolated point is solely influenced by the
nearby gridded points based on the inverse of the distance between the interpolated point and the
gridded points. The interpolations were carried out on an individual ecodistrict basis and were
based on both the number of precipitation-gauge stations and number of 0.5° x 0.5° grid cells
within the ecodistrict in question. For instance, when a single precipitation-gauge station was
located within an ecodistrict, the value of the interpolated point was calculated by using all of the
gridded points within that ecodistrict. When two or more precipitation-gauge stations were within
the same ecodistrict, their interpolated values were calculated by using the same numbers of
gridded points but with different weightings based on inverse distance. In the case when an
ecodistrict contained one grid cell, no weighting was used and the interpolated value was equal to
the nearest grid point.
4.2.  **Comparison of probability distributions using Kolmogorov-Smirnov test**
A two-sample, non-parametric Kolmogorov-Smirnov (K-S) test was used to compare the
cumulative distribution functions (CDFs) of gridded precipitation products with the AHCCD. The
null hypothesis ($H_0$) was that the two datasets came from same population. For each season,
monthly total precipitation data were used to avoid commonly known issues of numerous zero
values in the daily precipitation data that might affect significance. The K-S test was repeated
independently for all precipitation-gauge stations at 5 % significance level ($\alpha = 0.05$). A measure
of reliability (in percent) was calculated based on counting the number of stations that do not reject
the null hypothesis (any *p*-values greater than 0.05) over the total number of stations (145 and 137
stations in long-term and short-term comparison respectively), as shown in Eq. (1).
$\% \text{ of reliability} = \frac{number\ of\ stations\ that\ support\ H_0}{total\ number\ of\ precipitation\ gauge\ stations} \cdot 100$           (1)
4.3.  **Comparison of gridded precipitation data using performance measures**
Since the generation of the climate model-based precipitation products (PCIC dataset and NA-
CORDEX dataset) only preserved the statistical properties without considering the day-by-day
sequencing of precipitation events in the observational record, these two datasets were excluded
from the following comparison, which only focused on the station-based and reanalysis-based
gridded products. In particular, these products were assessed in their ability to represent the daily
variability of precipitation amounts in different ecozones by four performance measures:
percentage of bias ($PBias$) ($P_{Bias}$), root-mean-square-error ($RMSE$) ($E_{rms}$), correlation coefficient
($r$), and standard deviation ratio ($\sigma_G/\sigma_R$), as shown by Eqs. (2) to (5), respectively.
$$P_{Bias;s} = \frac{\sum_i^N (G_i - R_i)}{\sum_i^N (R_i)} \cdot 100 \tag{2}$$
$$E_{rms;s} = \sqrt{\frac{\sum_i^N (G_i - R_i)^2}{N}} \tag{3}$$
$$r_s = \frac{\sum_i^N (G_i - \bar{G})(R_i - \bar{R})}{\sqrt{\sum_i^N (G_i - \bar{G})^2}\sqrt{\sum_i^N (R_i - \bar{R})^2}} \tag{4}$$
$$(\sigma_G/\sigma_R)_s = \frac{\sqrt{\frac{\sum_i^N (G_i - \bar{G})^2}{N}}}{\sqrt{\frac{\sum_i^N (R_i - \bar{R})^2}{N}}} \tag{5}$$
where $s$ is the season, $G$ and $R$ are the spatial average of the daily gridded precipitation product
and the reference observation dataset (precipitation-gauge stations) respectively, $\bar{G}$ and $\bar{R}$ are the
daily mean of gridded precipitation product and point station data over the time spans (1979-2012
and 2002-2012), respectively, $i$ is the $i$-th day of the season, and $N$ is the total numbers of day in
the season. These four performance measures examined different aspects of the gridded
precipitation products, with $PBias$ for accuracy of product estimation, $RMSE$ for magnitude of
the errors, $r$ for strength and direction of the linear relationship between gridded products and
precipitation-gauge station data, and $\sigma_G/\sigma_R$ for amplitude of the variations.
5.    **Results**
5.1.    **Reliability of precipitation products**
The percentage of reliability of each precipitation dataset during every season for the periods of
1979 to 2012 and 2002 to 2012 across Canada is shown in Fig. 2. The higher the percentage, the
more reliable the precipitation dataset in question. In general, for long-term comparison (Fig. 2
left panel), WFDEI [GPCC] provided the highest percentage of reliability for the individual
seasons (from spring to winter: 72.5 %, 81.4 %, 70.3 %, and 50.3 %) while NARR had the lowest
percentage (24.8 %, 45.5 %, 27.6 %, and 11.7 %). Therefore in spring, WFDEI [GPCC] is not
significantly different for 72.5 % of the 145 precipitation-gauge stations while for NARR it is only
24.8 %. ANUSPLIN is second in spring and summer (56.6 % and 73.1 %) and WFDEI [CRU] in
autumn and winter (63.4 % and 45.5 %).
Regarding the PCIC ensembles, the different GCMs provided a range of reliabilities for the
individual seasons. MPI-ESM-LR performed the best in summer (70.2 %) and CanESM2 in
autumn (45.5 %). GFDL-ESM2G generally gave more reliable estimates in spring and winter (57.4
% and 41.7 %). Overall, the performance of MPI-ESM-LR (52.0 %) was the best among the GCMs,
followed by GFDL-ESM2G (50.1 %), CanESM2 (47.8 %), and HadGEM2 (36.2 %).  In terms of
statistical downscaling methods, the BCCAQ was on average slightly better than BCSD (49.5 %
versus 44.0 %) with the former having a greater similarity in spring and summer as opposed to
autumn and winter. These small differences therefore suggest that both methods are similar. With
respect to the NA-CORDEX ensembles, the CRCM5 RCM gave the most reliable estimates in
summer and autumn regardless of the GCM used. CanRCM4 had the best reliability in spring (49.4
%) whereas RegCM4 had the poorest reliability in spring and summer (24.4 % and 34.0 %).
Overall, the reliability of MPI-ESM-LR (44.7 %) was better than that of CanESM2 (42.5 %)
regardless of the RCMs used whereas the reliability of CRCM5 (43.6 %) was the best among the
RCMs, followed by CanRCM4 (41.2 %), and RegCM4 (32.5 %). It should also be noted that in
all cases, the gridded station-based and reanalysis-based products outperformed the climate model-
simulated products.
With regard to the short-term comparison (Fig. 2 middle panel), ANUSPLIN showed better
performance in summer with 94.1 % of reliability among the 137 precipitation-gauge stations
while CaPA indicated better skill in winter with 68.6 % of reliability. Again, WFDEI [GPCC] in
general provided the most consistent and reliable estimates with over 65 % of reliability in all four
seasons. It is interesting to note that for the most part, there is a higher percentage of reliability in
short-term period compared to long-term period. Reasons for this are not clear but can be partly
attributed to the fact that the power of K-S test (i.e. the probability of rejecting the null hypothesis
when the alternative is true) decreases with the number of samples.
Figures 3, 4 and 5 display the seasonal distributions of *p*-values using the K-S test in the 15
ecozones for long-term and short-term comparison, respectively. Due to the uneven distribution of
precipitation-gauge stations across Canada, the number of stations in each ecozone are different
(Table 4), with no stations in Region 1 (Arctic Cordillera), and Regions 2 to 5, 10, 12, and 15 have
less than 10 stations. The percentage of missing values in precipitation-gauge station in Region 11
exceeded 10 % in the period of 2002 to 2012 and thus Region 11 was excluded in the short-term
comparison. As a result, two representations were used to show the distributions of *p*-values.
Regions having more than or equal to 10 stations (6 to 9 and 13, 14) were shown in box-whisker
plots. Regions having less than 10 stations are indicated by hollow circles with each representing
one *p*-value at one precipitation-gauge station. Different colours in the figures corresponded to the
various precipitation products. The higher the number of high *p*-values ($> 0.05$) in each ecozone
(either represented by a cluster of hollow circles or a thick black line in box-whisker plots towards
1 in y-axis in Figs. 3, 4 and 5), the more confidence (more consistent) of each gridded precipitation
datasets in that ecozone.
From 1979 to 2012 (Fig. 3), in regions where more precipitation-gauge stations were available (6
to 10, 13, and 14), the consistency of each type of precipitation products is explored by assessing
the median of the *p*-values. Overall, all the precipitation products showed very low reliability and
consistency in winter among these ecozones and in every season in Regions 13 and 14 (Pacific
Maritime and Montane Cordillera) as the medians were close to zero, despite a couple of locations
having higher chance of same CDFs as in the precipitation-gauge station data. The WFDEI [GPCC]
dataset provided the highest consistency in the remaining three seasons except for Region 7
(Atlantic Maritime) where ANUSPLIN showed higher medians (0.51 and 0.46) than WFDEI
[GPCC] (0.42 and 0.42) in spring and autumn respectively. Noticeably NARR provided the lowest
median among the reanalysis-based datasets in all four seasons in Regions 6 to 8 but gave fairly
consistent estimates in Regions 9 and 10, especially in summer in Region 9 (Boreal Plain) where
it came second after WFDEI [GPCC]. The medians of Princeton were similar with those of
ANUSPLIN on average in these regions except for summer in which ANUSPLIN offered higher
medians than Princeton. WFDEI [CRU] generally showed consistent estimates among these
ecozones with medians well above 0.05 except for Region 7 (Atlantic Maritime) in spring and
autumn. From 1979 to 2005 (Fig. 5), the PCIC ensembles and the NA-CORDEX ensembles
showed different degrees of consistency among their GCM members with generally higher $p$-
values using BCCAQ method than BCSD method in spring and summer regardless of GCMs in
the PCIC datasets, whereas CanESM2 was generally having higher consistency and reliable
estimates than MPI-ESM-LR in spring and summer but opposite case in autumn in the NA-
CORDEX ensembles.
In ecozones above 60° N (Regions 2 to 5, 11, and 12), almost all the precipitation products had
lower chance of having same CDFs as the precipitation-gauge stations, especially in spring,
autumn, and winter in Region 3 (Southern Arctic) and spring and summer in Region 11 (Taiga
Cordillera). The WFDEI [GPCC] and WFDEI [CRU] generally tended to provide higher $p$-values
in these regions in spring and summer, followed by the NARR dataset. The NA-CORDEX
ensembles provided slightly higher chance of having same CDFs as the precipitation-gauge
stations than the PCIC ensembles in Regions 2 to 5 in spring and autumn whereas the opposite
case was shown in Region 12 (Boreal Cordillera) in spring.
For the shorter time period of 2002 to 2012 (Fig. 4), CaPA showed the highest consistency in
winter in Regions 6, 8, 9, and 13 whereas ANUSPLIN was the highest in summer in Regions 8,
13, and 14, echoing the results found in Fig. 2. However, the reliability and consistency of CaPA
in summer was not particularly high, especially in Regions 8 and 13 where the medians were
approaching zero. In addition, in ecozones above 60° N, the performances of CaPA were generally
similar to that of the WFDEI [GPCC] with higher chance of providing reliable estimates in autumn.
Similar performances were seen among the other precipitation products in the period of 2002 to
2012 as compared with the long-term performance.
5.2. **Daily variability of precipitation (station- and reanalysis-based products)**
The accuracy ($PBias$), magnitude of the errors ($RMSE$), strength and direction of the relationship
between gridded products and precipitation-gauge station data ($r$), and amplitude of the variations
($\sigma_G/\sigma_R$) are shown in Figs 6 and 7 for the period of 1979 to 2012 and 2002 to 2012, respectively.
In general, the gridded precipitation products that agree well with the precipitation-gauge station
data should have relatively high correlation and low RMSE, low bias and similar standard
deviation (light grey or dark grey squares in Figs. 5 and 6).
With respect to long-term comparison, in terms of overall accuracy among the four seasons,
ANUSPLIN performed relatively better in Region 11 (Taiga Cordillera) with smallest positive
$PBias$ (+0.5 %) while the rest of the gridded products had negative $PBias$ ranging from -1.4 %
(NARR) to -67.6 % (Princeton). However, ANUSPLIN was associated with a generally negative
$PBias$ for the rest of the ecozones ranging from -5.3 % (Region 13 Pacific Maritime) to -29.6 %
(Region 3 Southern Arctic), except for Regions 12 (Boreal Cordillera) and 14 (Montane
Cordillera). On the other hand, WFDEI [CRU] and WFDEI [GPCC] had similar performances
across different regions except in spring when the former underestimated the precipitation amounts
by 63.0 % but the latter overestimated by 5.3 % in Region 11 (Taiga Cordillera). Differences could
also be found in Region 7 (Atlantic Maritime) where WFDEI [CRU] overestimated precipitation
amounts in spring, autumn, and winter by 10.6 %, 7.1 %, and 7.5 % while the accuracy of WFDEI
[GPCC] was within -3.5 % to 0.5 % and it was the opposite case in Region 12 (Boreal Cordillera)
in autumn and winter. With the exception of Regions 13 and 14, Princeton generally provided the
overall largest underestimation of precipitation amounts across different ecozones by -25.9 %, -
24.8 %, and -34.6 % in spring, autumn, and winter respectively. NARR performed second worst
in spring (-19.0 %), autumn (-20.3 %), and winter (-27.1 %) and first in summer (-18.1 %). In
general, all gridded products tended to overestimate total precipitation in Regions 12 to 14, while
Region 14 (Montane Cordillera) had the overall highest positive $PBias$ ranging from 17.1 %
(WFDEI [GPCC]) to 44.2 % (WFDEI [CRU]).
When examining the magnitude of errors, ANUSPLIN showed generally better correspondence
with precipitation-gauge station data, providing the overall lowest $RMSE$ across ecozones in four
seasons (2.50 mm/day, 3.24 mm/day, 2.79 mm/day, and 2.45 mm/day) with the only exception in
spring in Region 15 (Hudson Plain). Moreover, ANUSPLIN had the overall highest $r$ across
ecozones in four seasons (0.75, 0.78, 0.80, and 0.74). On the contrary, Princeton had the worst
performance in both magnitude of errors and correlation with observations irrespective of ecozone
or season, with the grand $RMSE$ and $r$ of 5.65 mm/day and 0.17 respectively. The performances
of WFDEI [CRU], WFDEI [GPCC], and NARR were in between ANUSPLIN and Princeton and
they shared similar $RMSE$ and $r$ across different regions and seasons, with very high magnitude

of errors in Regions 6 to 8, and 13 and fair correlation in Regions 6 to 14 and minor regional and seasonal differences. The resulting values of the $RMSE$ metric in Regions 7 (Atlantic Maritime) and 13 (Pacific Maritime) tended to be larger than that of other ecozones. However, the other metrics such as $PBias$ and $r$ showed better performance in these regions. This suggests that higher $RMSE$ values can be mainly attributed to the fact that precipitation amounts are higher in the maritime regions.

Regarding the amplitude of variations, NARR had the lowest variability across different regions in all four seasons (0.70, 0.67, 0.68, and 0.60), followed by ANUSPLIN (0.84, 0.77, 0.76, and 0.75). WFDEI [GPCC] had the most similar standard deviations as that of precipitation-gauge station data in Regions 5 to 8, 13, and 14 in autumn and winter, while WFDEI [CRU] had about the same standard deviations in Regions 6 to 8 in autumn only. Unlike ANUSPLIN and NARR which were consistently having too little variability across different ecozones, Princeton estimated the amplitude of variations with more diversified regional and seasonal patterns. Princeton estimated $\sigma_G/\sigma_R$ the best in Regions 4 to 10 in summer and Regions 9, 10, and 12 in autumn. However, the dataset had variations that were much larger than precipitation-gauge station data in Regions 7 and 8 in four seasons except summer, Region 13 in four seasons except winter, Region 14 in all seasons but too little variability in Regions 3, 11, and 15 in all seasons.

Concerning the short-term comparison, the performance of CaPA generally resembled that of ANUSPLIN in terms of accuracy, with general underestimation of precipitation amounts in Regions 4 to 10 in four seasons and overestimation in Region 12 and 13 especially in spring. CaPA had similar overestimation in Region 14 (Montane Cordillera) in winter as the rest of the gridded products but performed the best in estimating the precipitation amounts in other seasons of the region. CaPA also performed the best in Regions 5 and 15 in autumn among the gridded precipitation products. However, while all the gridded products experienced negative $PBias$ in Region 3 (Southern Arctic) in summer, CaPA performed the opposite with a positive $PBias$ of 10.8 %. Similar to ANUSPLIN, CaPA had the second lowest overall $RMSE$ (2.70 mm/day, 3.74 mm/day, 3.35 mm/day, and 3.05 mm/day) and $r$ (0.72, 0.73, 0.75, and 0.70) across ecozones in all seasons, respectively. Despite its better performances in terms of $RMSE$ and $r$, CaPA was generally not able to capture satisfactorily the amplitude of variations, with consistently lower values across different regions for seasons (0.83, 0.82, 0.85, and 0.72). In terms of $\sigma_G/\sigma_R$, CaPA

showed more skill compared to ANUSPLIN (0.72, 0.76, 0.74, and 0.64) and NARR (0.75, 0.75,
0.72, and 0.63).
Some regional and seasonal differences were observed in the other gridded precipitation products.
For instance, seasonally, WFDEI [CRU] performed well in Region 8 (Mixedwood Plain) as judged
by low $PBias$ (-1.7 % to 4.3 %) for the period of 1979 to 2012 but showed higher positive $PBias$
in autumn and winter (7.1 % and 5.3 %) for the period of 2002 to 2012. WFDEI [GPCC] also had
higher positive $PBias$ in Region 2 (Northern Arctic) in summer (7.4 % as compared to 1.2 %) and
winter (33.3 % as compared to 9.9 %). In terms of magnitude of errors and correlation with
observations, the five gridded products in the long-term comparison performed similarly in the
period of 2002 to 2012, with ANUSPLIN having the lowest grand $RMSE$ and $r$ of 2.88 mm/day
and 0.78 and Princeton being the worst again with the highest grand $RMSE$ and $r$ of 6.12 mm/day
and 0.16 respectively. Equally, the performances of ANUSPLIN and NARR in capturing the
amplitude of variations were again consistently having too little variability across different
ecozones. Princeton also demonstrated similar regional and seasonal differences as in the long-
term comparison with higher variability in Regions 6 to 8 in all seasons except summer. WFDEI
[CRU] and WFDEI [GPCC] both performed well in Regions 6 to 8, 12, and 14 in autumn.
6.    **Discussion**
The preceding has provided insight into the relative performance of various gridded precipitation
products over Canada relative to gauge measurements over different seasons and ecozones. Results
showed that there is no particular product that is superior for all performance measures although
some datasets are consistently better. Based on the performances, one could broadly characterize
the station- and reanalysis-based precipitation products into four groups: (1) ANUSPLIN and
CaPA with negative $PBias$, low $RMSE$, high $r$, and small $\sigma_G/\sigma_R$; (2) WFDEI [CRU] and WFDEI
[GPCC], with relatively small $PBias$, high $RMSE$, fair $r$, and similar standard deviation; (3)
Princeton, with negative $PBias$, high $RMSE$, low $r$, and a mixture of large and small $\sigma_G/\sigma_R$; and
(4) NARR, with negative $PBias$, high $RMSE$, fair $r$, and small $\sigma_G/\sigma_R$. Among the reanalysis-
based gridded products, Princeton performed the worst in all seasons and regions in terms of
minimizing error magnitudes (Figs. 8 and 9). Princeton was especially poor in winter (Fig. 8) and
showed significant underestimation in regions above 60° N (Fig. 9). This could be due to the use
of the NCEP-NCAR reanalysis as the basis to generate the dataset, which have been shown to be
less accurate than NCEP-DOE reanalysis (used in NARR) and ERA-40 reanalysis (used in WFD)
(Sheffield et al., 2006). The better performance of NARR in capturing the timings and amounts of
precipitation compared to Princeton was probably because NCEP-DOE reanalysis was a major
improvement upon the earlier NCEP-NCAR reanalysis in both resolution and accuracy. However,
the overall reliability of NARR was among the poorest mainly because of non-assimilation of
gauge precipitation observations over Canada from 2004 onwards, as reported by Mesinger et al.
(2006). ANUSPLIN and CaPA performed well in capturing the timings and minimizing the error
magnitudes of the precipitation, despite their general underestimation across Canada ($PBias$
ranging from -7.7 % (Region 13) to -40.7 % (Region 3) and -2.0 % (Region 15) to -17.1 % (Region
8) in the period of 2002 to 2012) (Fig. 9) and too little variability (grand $\sigma_G/\sigma_R$ of 0.72 and 0.80
of the same period). This was not surprising given that the generation of the products was based
on the unadjusted precipitation-gauge stations where the total rainfall amounts were increased after
adjustment (Mekis and Vincent, 2011). WFDEI [CRU] and WFDEI [GPCC], on the other hand,
performed well in estimating the accuracy and amplitude of variations, but not the timings and
error magnitudes of the precipitation. This could probably due to the positive bias offsetting the
negative bias resulting in small mean bias, but was picked up by $RMSE$ that gives more weights
to the larger errors. The larger errors could result from a mismatch of occurrence of precipitation
in the time series, as reflected by the fair correlation coefficients (grand $r$ of 0.52 and 0.50 for
WFDEI [CRU], 0.54 and 0.53 for WFDEI [GPCC], for time periods of 1979 to 2012 and 2002 to
2012 respectively).
By matching the statistical properties of the adjusted gauge measurements at monthly time scale,
one could establish the confidence in using the climate model-simulated products for long-term
hydro-climatic studies. Comparing the overall reliability of the PCIC and NA-CORDEX datasets,
it was found that for the individual seasons the PCIC ensembles (spring, summer, and winter: 54.0
%, 64.7 %, and 35.7 %) outperformed the NA-CORDEX ensembles (39.1 %, 45.0 %, and 31.3 %)
except in autumn when the NA-CORDEX ensembles (45.5 %) provided slightly higher reliability
than the PCIC ensembles (45.2 %). The better reliability of the PCIC datasets could be due to the
use of ANUSPLIN to train the GCMs and thus, the statistical properties of the downscaled outputs
are guided by those of the ANUSPLIN. Similarly, for ecozones where more than 10 precipitation-
gauge stations could be found (Regions 6 to 9, 13 and 14), the PCIC ensembles (reliability ranging
from 35.7 % to 64.4 %) also outperformed the NA-CORDEX ensembles (from17.2 % to 61.6 %).
This would suggest that the PCIC ensembles may be the preferred choice for long-term climate
change impact assessment over Canada, although further research is required.
The evaluations of this comparison were impacted by the spatial distribution of adjusted
precipitation-gauge stations (Mekis and Vincent, 2011), which were assumed to be the best
representation of reality owing to efforts in improving the raw archive of the precipitation-gauge
stations. However, the major limitation of this dataset was the number of precipitation-gauge
stations that could be used for comparison. As aforementioned, due to temporal coverage not
encompassing the entire study period and not having a complete year of 2012, over half of the
precipitation-gauge stations were discarded from the analysis. Although the locations of the
remaining stations covered much of Canada, there are only one or a few stations located in some
of the ecozones (e.g. Region 3 to 5, 11, and 15). Even in Region 10 (Prairie) there are only nine
precipitation-gauge stations for analysis. While the reliability of different types of gridded products
could be tested in these ecozones, the consistency of the performance of each gridded product
could not be established due to small sample sizes.
In addition, results from the above analysis should be interpreted with care because the
precipitation-gauge station data are point measurements whereas the gridded precipitation
products are areal averages, of which the accuracy and precision of the estimates can be very
different given the non-linear responses of precipitation (Ebert et al., 2007). When comparing point
measurements and areal-average estimates, fundamental challenges occur because of the sampling
errors arising from different sampling schemes and errors related to gauge instrumentation
(Bowman, 2005). It is therefore difficult to have perfect spatial matching between point
measurements (gauge stations) and areal-averaged estimates (gridded products) (Sapiano and
Arkin, 2009;Hong et al., 2007). However, in the absence of a sufficiently dense precipitation gauge
network in Canada, the options for assessing different gridded products are limited. The only
gridded product that is basically representing areal averages of precipitation (via interpolation)
based on ground observations is ANUSPLIN. As aforementioned (see Sect. 3.2.1), this product
has its own limitations and may not be qualified to be considered as the "ground truth". Therefore,
ANUSPLIN is also included in the pool of gridded products to be evaluated. Notwithstanding the
issues, using the selected gauge measurements would remain the best way for the evaluation of the
multiple gridded products because the set of gauges used had been adjusted (e.g. for undercatch)
and are the most accurate source of information on precipitation in Canada (although small with
limited spatial coverage). Also, given that all the gridded products are compared against this
common set of station observations, it is assumed that the bias that the difference between point
and areal data introduces into the analysis is consistent for all the products. Therefore, given the
current data situation, the preceding methods could be used for comparing the performance of
different daily gridded precipitation products.
## 7.  Conclusion
A number of gridded climate products incorporating multiple sources of data have recently been
developed with the aim of providing better and more reliable measurements for climate and
hydrological studies. There is a pressing need for characterizing the quality and error
characteristics of various precipitation products and assessing how they perform at different spatial
and temporal scales. This is particularly important in light of the fact that these products are the
main driver of hydrological models in many regions, including Canadian watersheds where
precipitation-gauge network is typically limited and sparse. This study was conducted to inter-
compare several gridded precipitation products of their probability distributions and quantify the
spatial and temporal variability of the errors relative to station observations in Canada, so as to
provide some insights for potential users in selecting the products for their particular interests and
applications. Based on the above analysis, the following conclusions can be drawn:
- In general, all the products performed best in summer, followed by autumn, spring, and
winter in order of decreasing quality. The lower reliability in winter is likely the result of
difficulty in accurately capturing solid precipitation.
- Overall, WFDEI [GPCC] and CaPA performed best with respect to different performance
measures. WFDEI [GPCC], however, may be a better choice for long-term analyses as it
covers a longer historical period. ANUSPLIN and WFDEI [CRU] also performed
comparably, with considerably lower quality than WFDEI [GPCC] and CaPA. Princeton
and NARR demonstrated the lowest quality in terms of different performance measures.
- Station-based and reanalysis-based products tended to underestimate total precipitation
across Canada except in southwestern regions (Pacific Maritime and Montane Cordillera)
where the tendency was towards overestimation. This may be the due to the fact that the

majority of precipitation-gauge stations are located at lower altitudes which might not
accurately reflect areal precipitation due to topographic effect.

• In southern Canada, WFDEI [GPCC] and CaPA demonstrated their best performance in
the western cold interior (Boreal Plain, Prairie, Montane Cordillera) in terms of timing and
magnitude of daily precipitation.

• In northern Canada (above 60° N), the different products tended to moderately (ranging
from -0.6 % to -40.3 %) and in cases significantly (up to -60.3 % in Taiga Cordillera)
underestimate total precipitation, while reproducing the timing of daily precipitation rather
well. It should be noted that this assessment was based on only a limited number of
precipitation-gauges in the north.

• Comparing the climate model-simulated products, PCIC ensembles generally performed
better than NA-CORDEX ensembles in terms of reliability and consistency in four seasons
across Canada.

• In terms of statistical downscaling methods, the BCCAQ method was slightly more reliable
than the BCSD method across Canada on the annual basis.

• Regarding GCMs, MPI-ESM-LR provides the highest reliability, followed by GFDL-
ESM2G, CanESM2, and HadGEM2. With respect to RCMs, CRCM5 performed the best
regardless of the GCM used, followed by CanRCM4, and RegCM4.

The findings from this analysis provide additional information for potential users to draw
inferences about the relative performance of different gridded products. Although no clear-cut
product was shown to be superior, researchers/users can use this information for selecting or
excluding various datasets depending on their purpose of study. It is realized that this investigation
only focused on the daily time scale at a relatively coarse 0.5° x 0.5° resolution suitable for large-
scale hydro-climatic studies. Further research is thus required towards performance assessment of
various products with respect to precipitation extremes, which often have the greatest hydro-
climatic impacts. As new products become available, similar comparisons should be conducted to
assess their reliability.
**Acknowledgements**
The financial support from the Canada Excellence Research Chair in Water Security is gratefully
acknowledged. Thanks are due to Melissa Bukovsky and Katja Winger from the NA-CORDEX
modelling group for providing access to RegCM4 and CRCM5 data used in this study. The authors
are also grateful to the various organizations that made the datasets freely available to the scientific
community.

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

List of Tables

| Dataset | Full Name | Type | Spatial Resolution | Temporal Resolution | Duration | Coverage | Reference |
|---|---|---|---|---|---|---|---|
| ANUSPLIN | Australian National University Spline | Station-based Interpolated | 300 arc-second (~0.0833°/ ~10 km) | 24 hr | 1950 – 2013 | Canada | Hutchinson et al. (2009) |
| CaPA | Canadian Precipitation Analysis | Station-based Model-derived | 10 km (~0.0833°) | 6 hr | 2002 – 2014 | North America | Mahfouf et al. (2007) |
| Princeton | Global dataset at the Princeton University | Reanalysis-based multiple source | 0.5° (~50 km) | 3 hr | 1901 – 2012 | Global | Sheffield et al. (2006) |
| WFDEI [CRU] | Water and Global Change Forcing Data methodology applied to ERA-Interim [Climate Research Unit] | Reanalysis-based multiple source | 0.5° (~50 km) | 3 hr | 1979 – 2012 | Global | Weedon et al. (2014) |
| WFDEI [GPCC] | Water and Global Change Forcing Data methodology applied to ERA-Interim [Global Precipitation Climatology Centre] | Reanalysis-based multiple source | 0.5° (~50 km) | 3 hr | 1979 – 2012 | Global | Weedon et al. (2014) |
| NARR | North American Regional Reanalysis | Reanalysis-based multiple source | 32 km (0.3°) | 3 hr | 1979 – 2015 | North America | Mesinger et al. (2006) |
| PCIC | Pacific Climate Impacts Consortium | Station-driven GCM | 300 arc-second (~0.0833°/ ~10 km) | 24 hr | Historical: 1950 – 2005 Projected: 2006 – 2100 | Canada | Pacific Climate Impacts Consortium; University of Victoria (Jan 2014) |
| NA-CORDEX | North America COordinated Regional climate Downscaling EXperiment | GCM-driven RCM | 0.22° (25 km) | 3 hr | Historical: 1950 – 2005 Projected: 2006 – 2100 | North America | Giorgi et al. (2009) |

*Table 2 GCMs chosen in the Pacific Climate Impacts Consortium (PCIC) dataset.*

| PCIC | Full Name | Country | Statistical Downscaling Method |
|---|---|---|---|
| GFDL-ESM2G_BCCAQ | Geophysical Fluid Dynamics Laboratory Earth System Model 2G | USA | Bias Correction Constructed Analogues with Quantile mapping reordering |
| GFDL-ESM2G_BCSD | | | Bias Correction Spatial Disaggregation |
| HadGEM2-ES_BCCAQ | Hadley Global Environmental Model 2 – Earth System | UK | Bias Correction Constructed Analogues with Quantile mapping reordering |
| HadGEM2-ES_BCSD | | | Bias Correction Spatial Disaggregation |
| CanESM2_BCCAQ | Second generation Canadian Earth System Model | Canada | Bias Correction Constructed Analogues with Quantile mapping reordering |
| CanESM2_BCSD | | | Bias Correction Spatial Disaggregation |
| MPI-ESM-LR_ BCCAQ | Max-Planck-Institute Earth System Model running on low resolution | Germany | Bias Correction Constructed Analogues with Quantile mapping reordering |
| MPI-ESM-LR_ BCSD | | | Bias Correction Spatial Disaggregation |

*Table 3 GCMs-RCMs chosen in the North America COordinated Regional climate Downscaling EXperiment (NA-CORDEX) dataset.*

| NA-CORDEX | Full Name | |
|---|---|---|
| | Global Circulation Model (GCM) | Regional Climate Model (RCM) |
| CanESM2 – CanRCM4 | Second generation Canadian Earth System Model | Fourth generation Canadian Regional Climate Model |
| CanESM2 – CRCM5_UQAM | | Fifth generation Canadian Regional Climate Model |
| MPI-ESM-LR – CRCM5_UQAM | Max-Planck-Institute Earth System Model running on low resolution | |
| MPI-ESM-LR – RegCM4 | | Fourth generation Regional Climate Model |

*Table 4 Number of precipitation-gauge stations within each Ecozone.*

| | Region (Ecozone) | Number of Precipitation-gauge Stations | |
| --- | --- | --- | --- |
| | | 1979 – 2012 | 2002 – 2012 |
| 1 | Arctic Cordillera | 0 | 0 |
| 2 | Northern Arctic | 4 | 4 |
| 3 | Southern Arctic | 1 | 1 |
| 4 | Taiga Plain | 2 | 2 |
| 5 | Taiga Shield | 4 | 5 |
| 6 | Boreal Shield | 31 | 29 |
| 7 | Atlantic Maritime | 10 | 9 |
| 8 | Mixedwood Plain | 18 | 16 |
| 9 | Boreal Plain | 14 | 14 |
| 10 | Prairie | 9 | 7 |
| 11 | Taiga Cordillera | 1 | 0 |
| 12 | Boreal Cordillera | 6 | 6 |
| 13 | Pacific Maritime | 15 | 15 |
| 14 | Montane Cordillera | 28 | 26 |
| 15 | Hudson Plain | 2 | 3 |
| | Total | 145 | 137 |

List of Figures

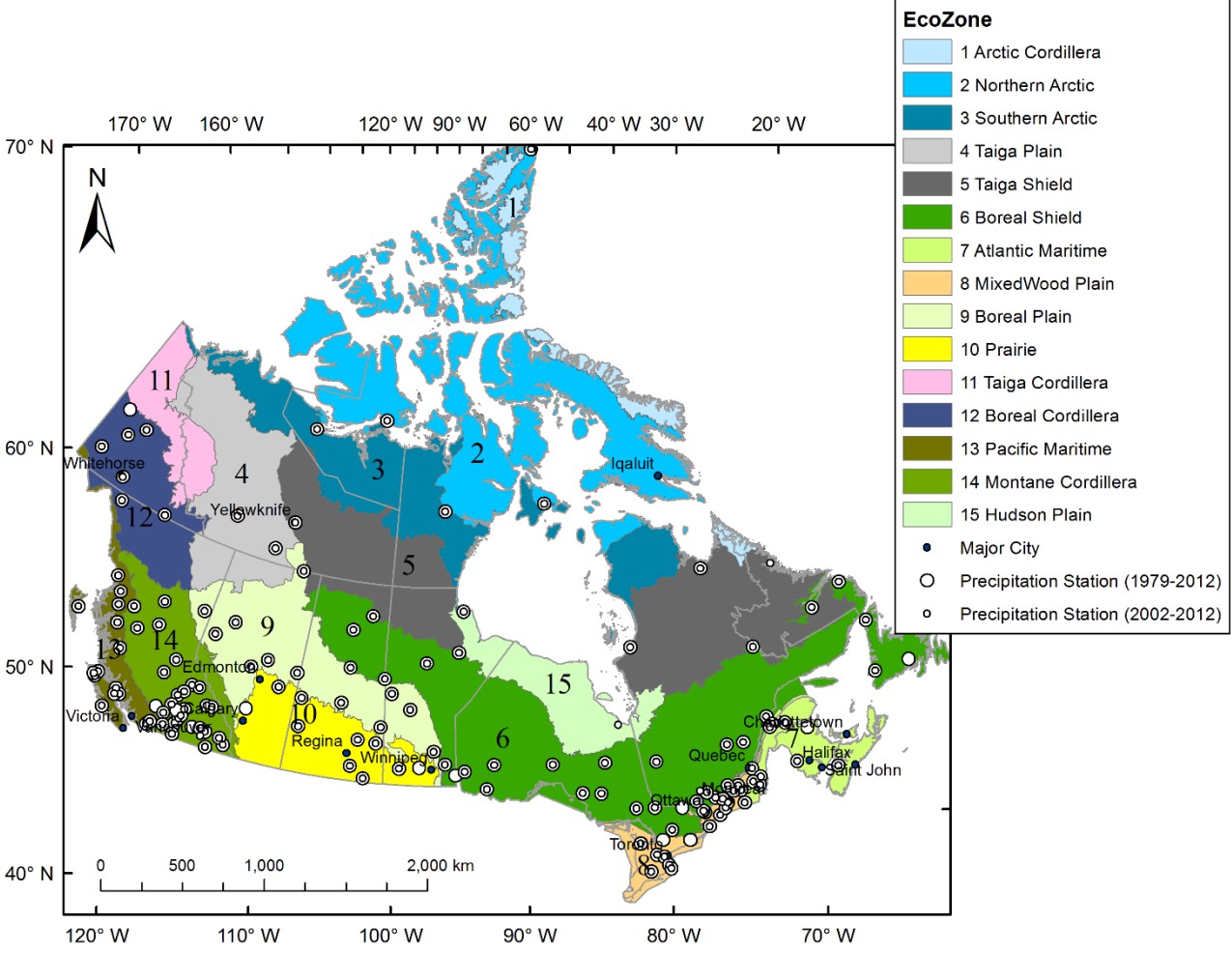

*Figure 1. 15 terrestrial ecozones of Canada with numerical codes indicating Region from 1 Arctic Cordillera to 15 Hudson Plain. Big (a total of 145) and small (a total of 137) white dots are the extracted precipitation-gauge stations from the Canadian adjusted and homogenized precipitation datasets of Mekis and Vincent (2011) for the period of 1979 to 2012 and 2002 to 2012 respectively. Black dots are major cities in Canada.*

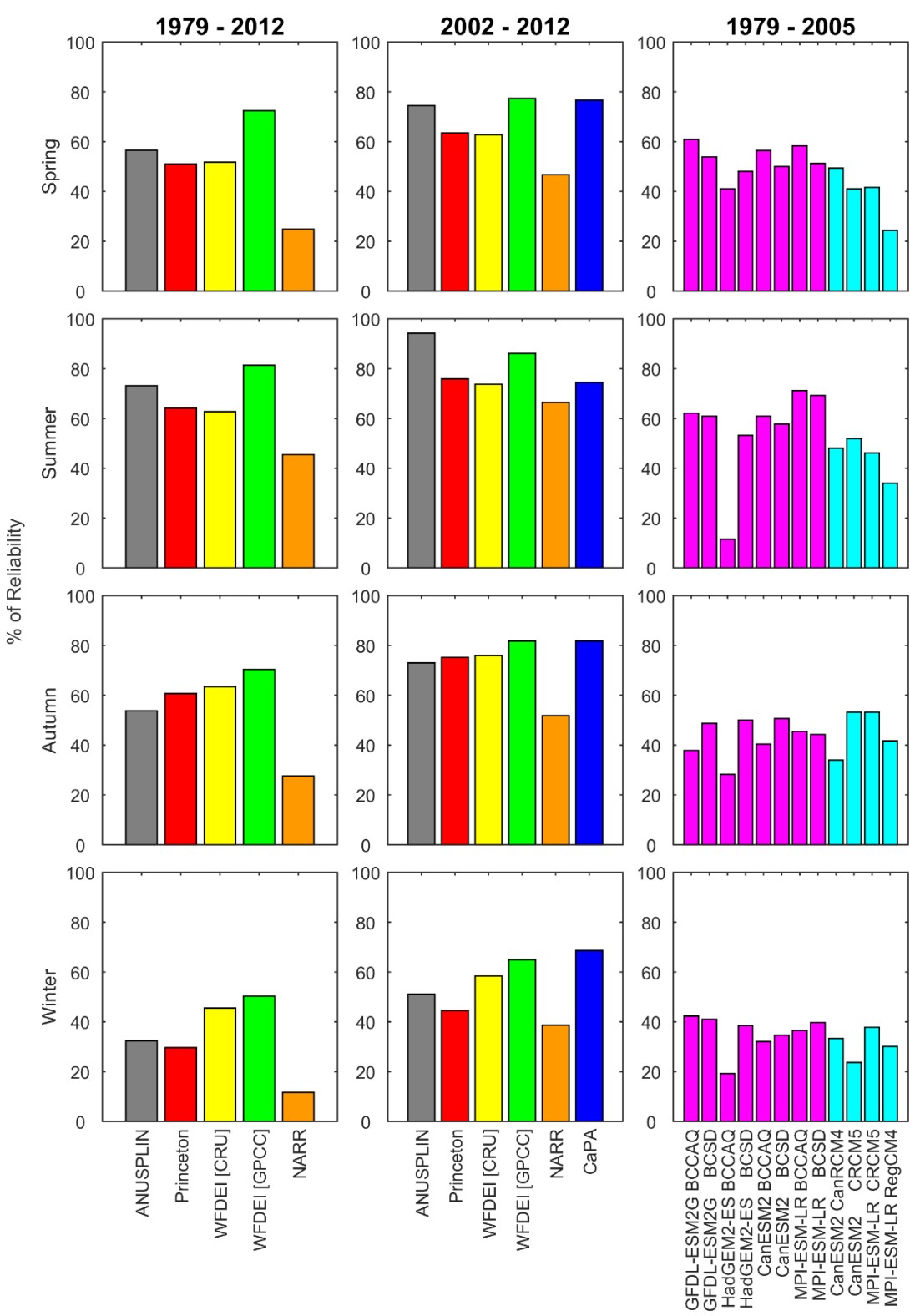

*Figure 2. The percentage of reliability, calculated by the Eq. (1), of each precipitation dataset in four seasons for the period of 1979 to 2012 (left panel), 2002 to 2012 (middle panel), and 1979 to 2005 (right panel) across Canada. The higher the percentage, the more reliable the precipitation dataset. Different colours represent different precipitation products, with magenta representing the whole PCIC datasets and cyan representing the whole NA-CORDEX datasets. The full names of the precipitation products are provided in Tables 1, 2, and 3.*

**1979 - 2012**

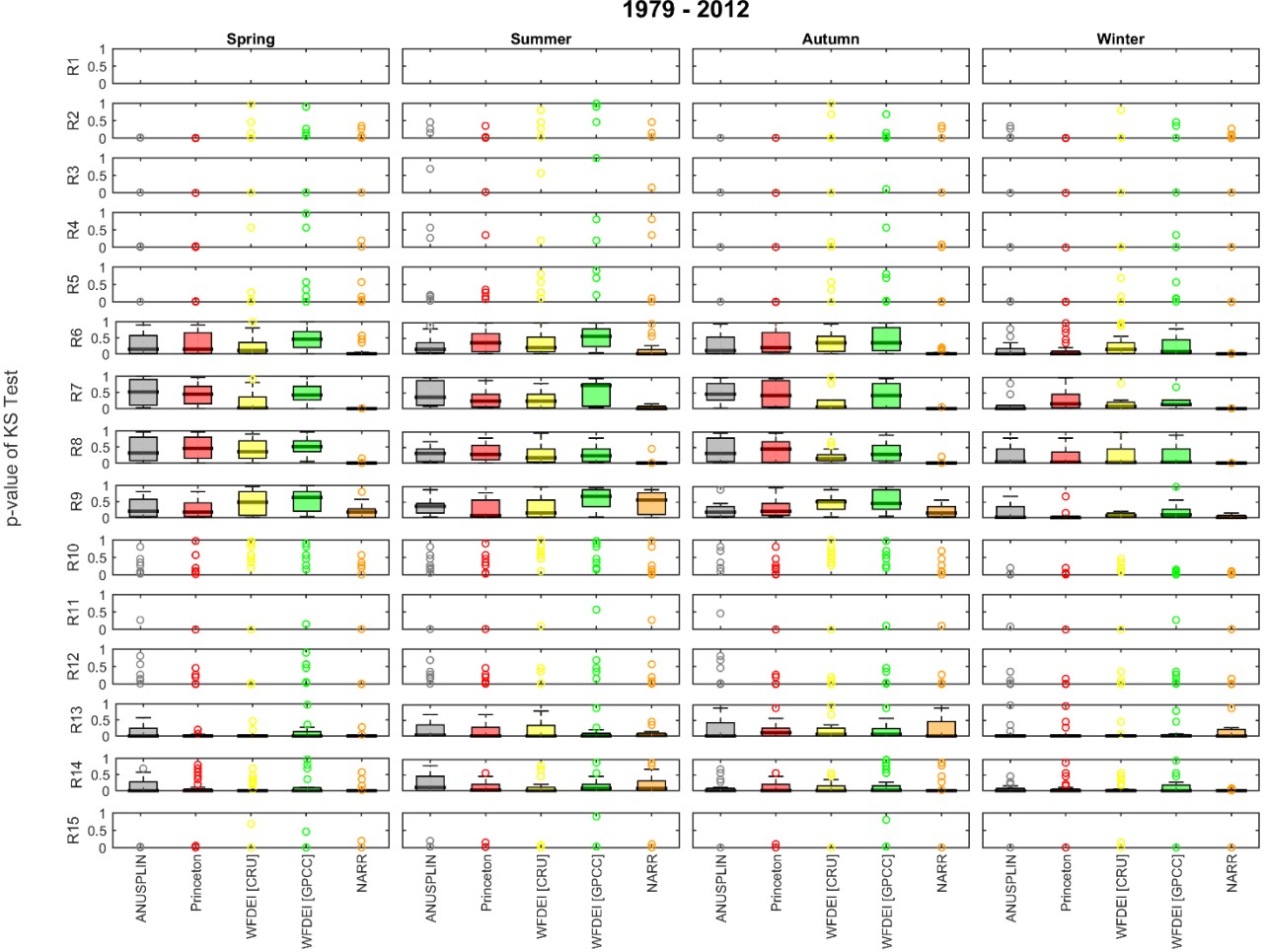

*Figure 3. Distributions of p-value of the K-S test in the 15 ecozones in four seasons for the period of 1979 to 2012 (long-term comparison without CaPA). Note that the numbers of precipitation-gauge stations in each ecozone are different (see Table 4). Each hollow circle represents one p-value of the K-S test conducted at one precipitation-gauge station, with no stations in Region 1 (R1). The p-values of Regions 6 to 9, and 13 to 14 (R6-R9, and R13-R14), which have more than or equal to 10 stations, were shown in box-whisker plots with bottom, band (black thick line) and top of the box indicating the 25th, 50th (median), and 75th percentiles, respectively.*

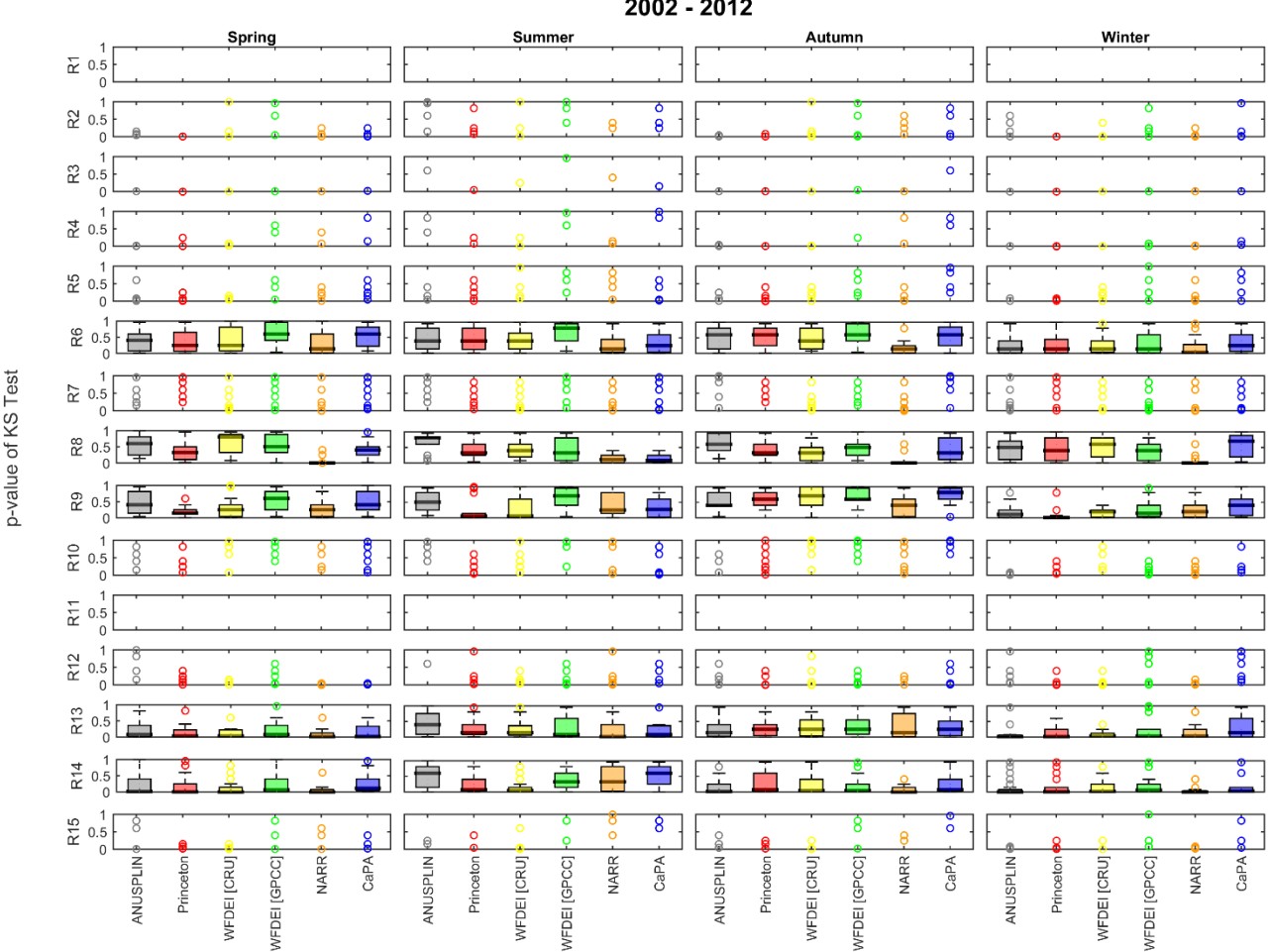

*Figure 4. Distributions of p-value of the K-S test in the 15 ecozones in four seasons for the period of 2002 to 2012 (short-term comparison with the inclusion of CaPA). Note that the numbers of precipitation-gauge stations in each ecozone are different (see Table 4). Each hollow circle represents one p-value of the K-S test conducted at one precipitation-gauge station. The percentage of missing values in precipitation-gauge station in Region 11 (R11) exceeded 10% and thus no K-S test was conducted. The p-values of Regions 6, 8 to 9, and 13 to 14 (R6, R8-R9, and R13-R14), which have more than or equal to 10 stations, were shown in box-whisker plots with bottom, band (black thick line) and top of the box indicating the 25th, 50th (median), and 75th percentiles, respectively.*

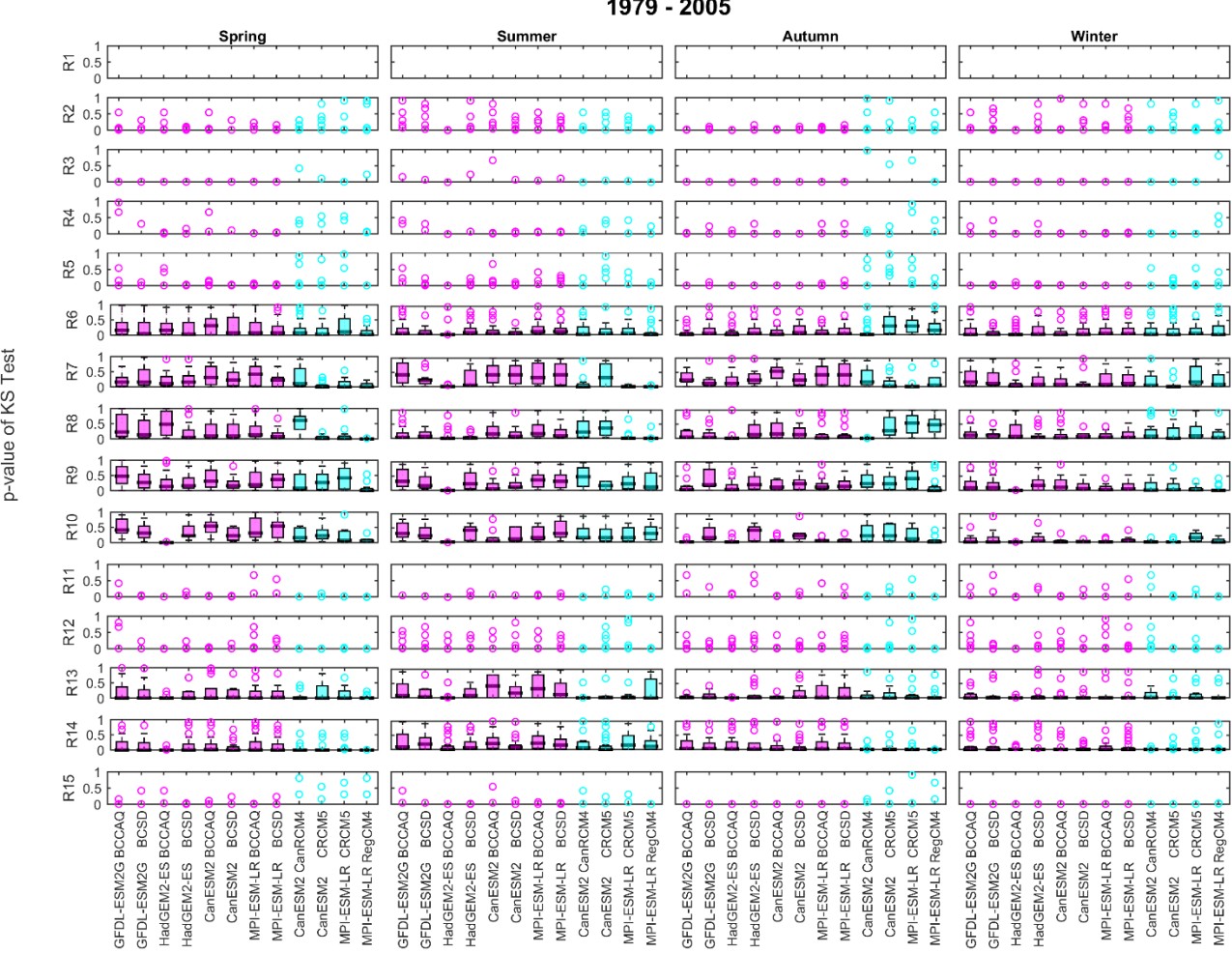

*Figure 5. Distributions of p-value of the K-S test in the 15 ecozones in four seasons for the period of 1979 to 2005 (long-term comparison of PCIC and NA-CORDEX). Note that the numbers of precipitation-gauge stations in each ecozone are different (see Table 4). Each hollow circle represents one p-value of the K-S test conducted at one precipitation-gauge station, with no stations in Region 1 (R1). The p-values of Regions 6 to 9, and 13 to 14 (R6-R9, and R13-R14), which have more than or equal to 10 stations, were shown in box-whisker plots with bottom, band (black thick line) and top of the box indicating the 25th, 50th (median), and 75th percentiles, respectively.*

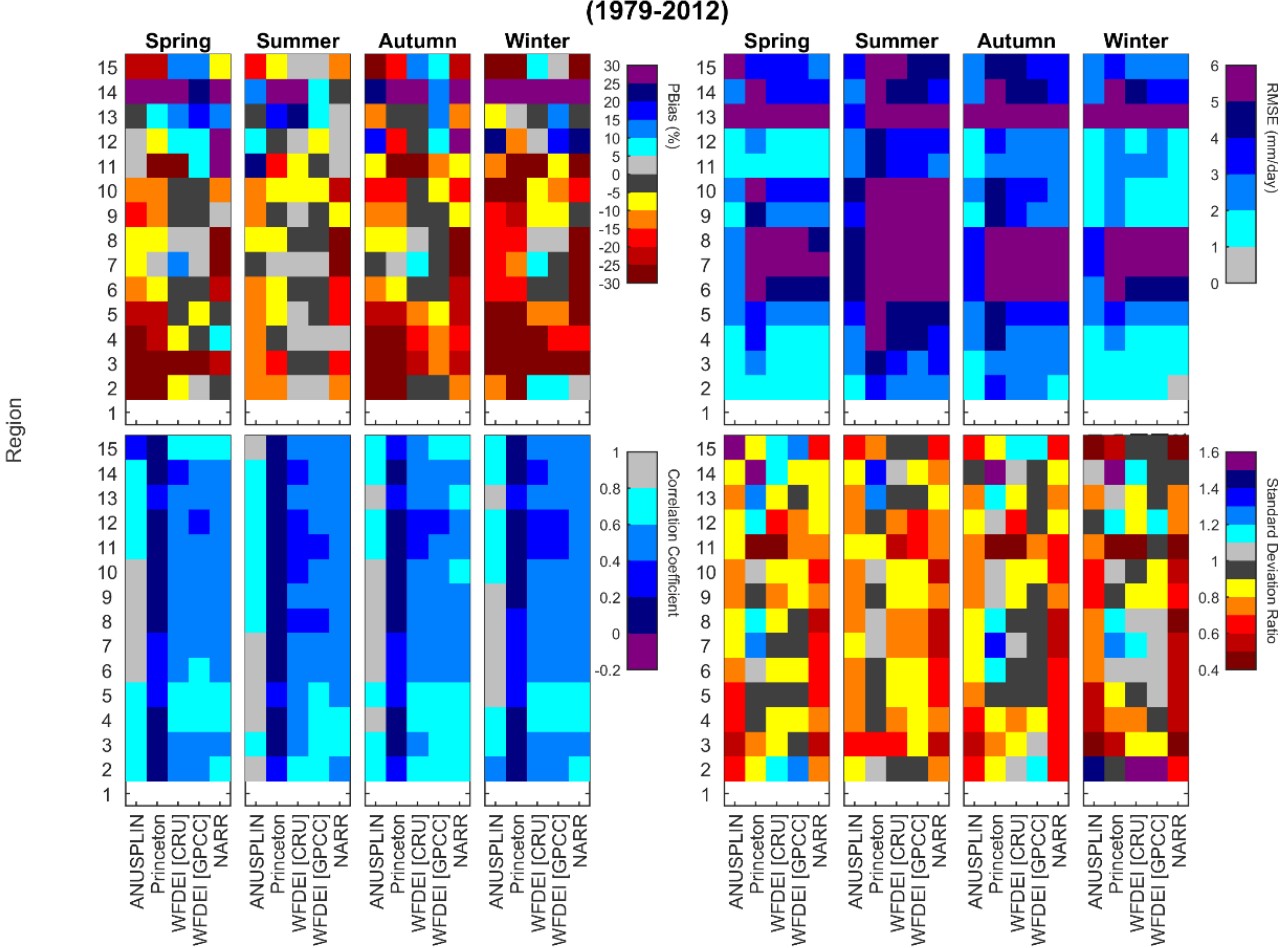

*Figure 6. Portrait diagram showing the accuracy (PBias) (top left), magnitude of the errors (RMSE) (top right), strength and direction of relationship between gridded products and precipitation-gauge stations (r) (bottom left), and amplitude of the variations ($\sigma_G/\sigma_R$) (bottom right) of each type of gridded precipitaiton products when evaluating against the precipitation-gauge station data in each ecozone (Region 1 to 15) in four seasons for the time period of 1979 to 2012. Each column indicates one gridded precipitation product and each row represents one ecozone with numerical code corresponding to region shown in Fig. 1. White indicates that no data are available due to no precipitation-gauge stations exisiting in that region.*

**(2002-2012)**

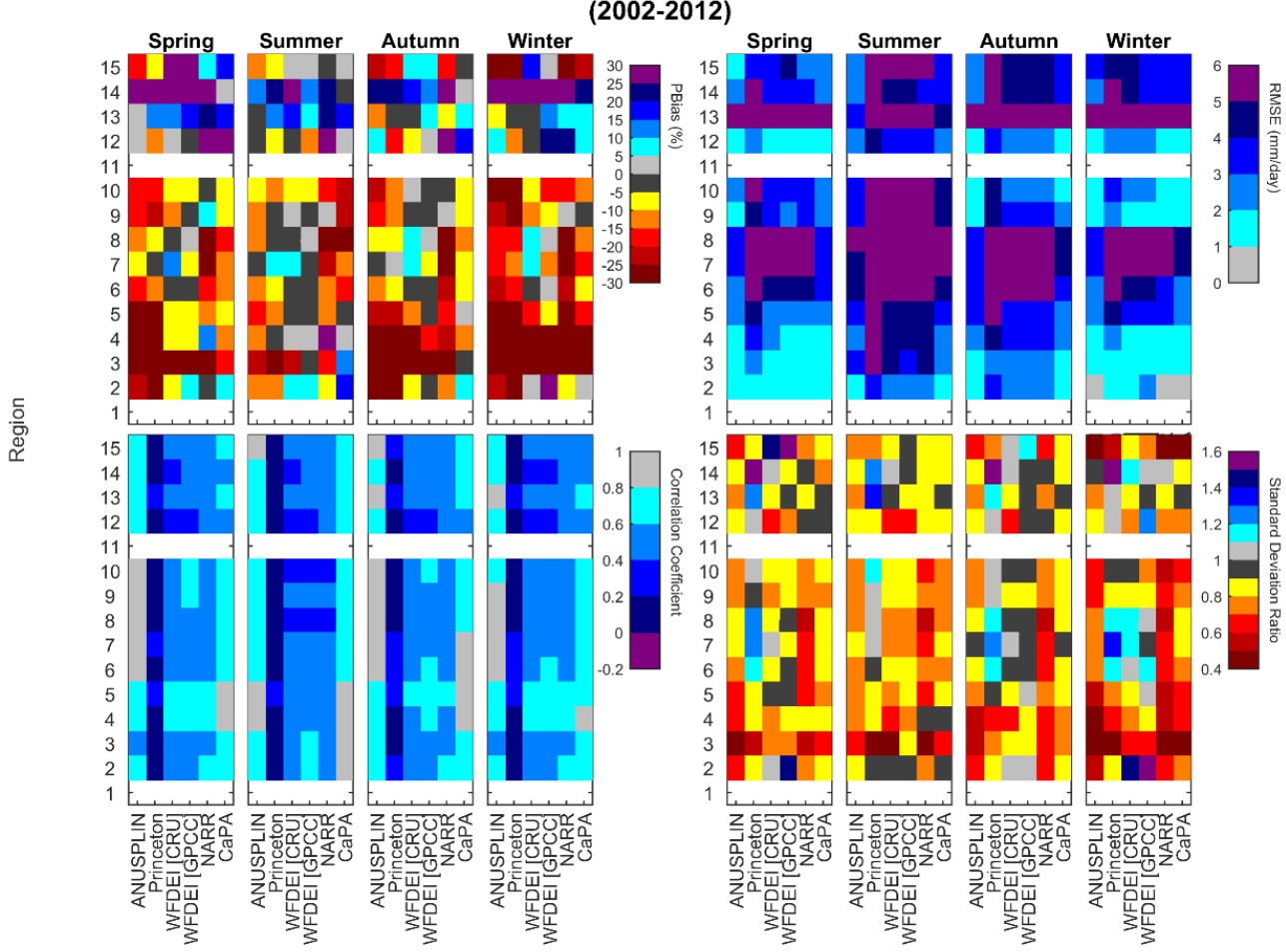

*Figure 7. Portrait diagram showing the accuracy (PBias) (top left), magnitude of the errors (RMSE) (top right), strength and direction of relationship between gridded products and precipitation-gauge stations (r) (bottom left), and amplitude of the variations ($\sigma_G / \sigma_R$) (bottom right) of each type of gridded precipitaiton products when evaluating against the precipitation-gauge station data in each ecozone (Region 1 to 15) in four seasons for the time period of 2002 to 2012. Each column indicates one gridded precipitation product and each row represents one ecozone with numerical code corresponding to region shown in Fig. 1. White indicates that no data are available due to no precipitation-gauge stations exisiting in that region.*

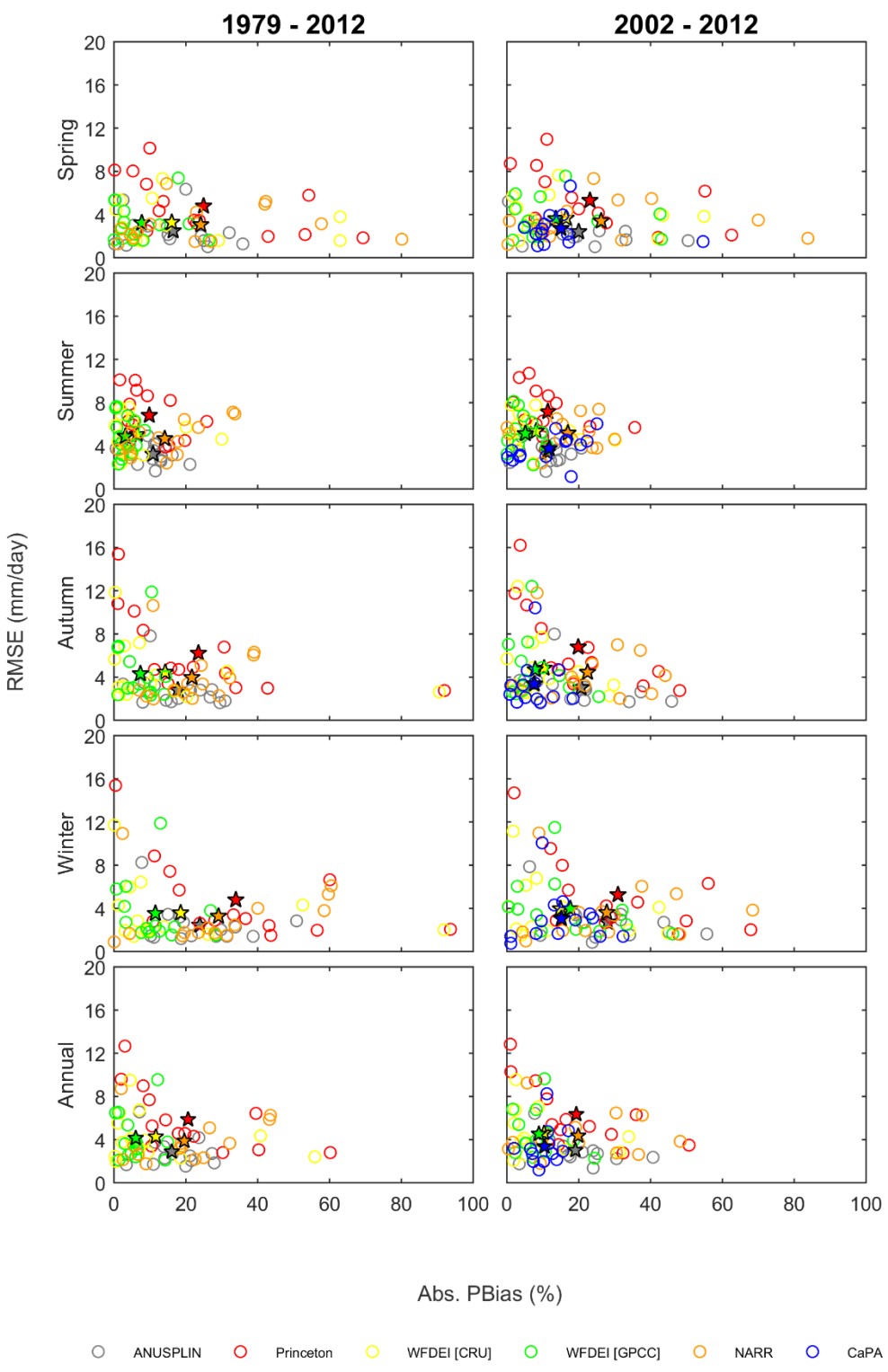

*Figure 8. Scatter plots showing absolute PBias (x-axis) versus RMSE (y-axis) of each precipitation dataset in four seasons and the entire year for the period of 1979 to 2012 (left panel) and 2002 to 2012 (right panel). Each hollow circle represents one ecozone and the solid stars indicate the overall average across ecozones.*

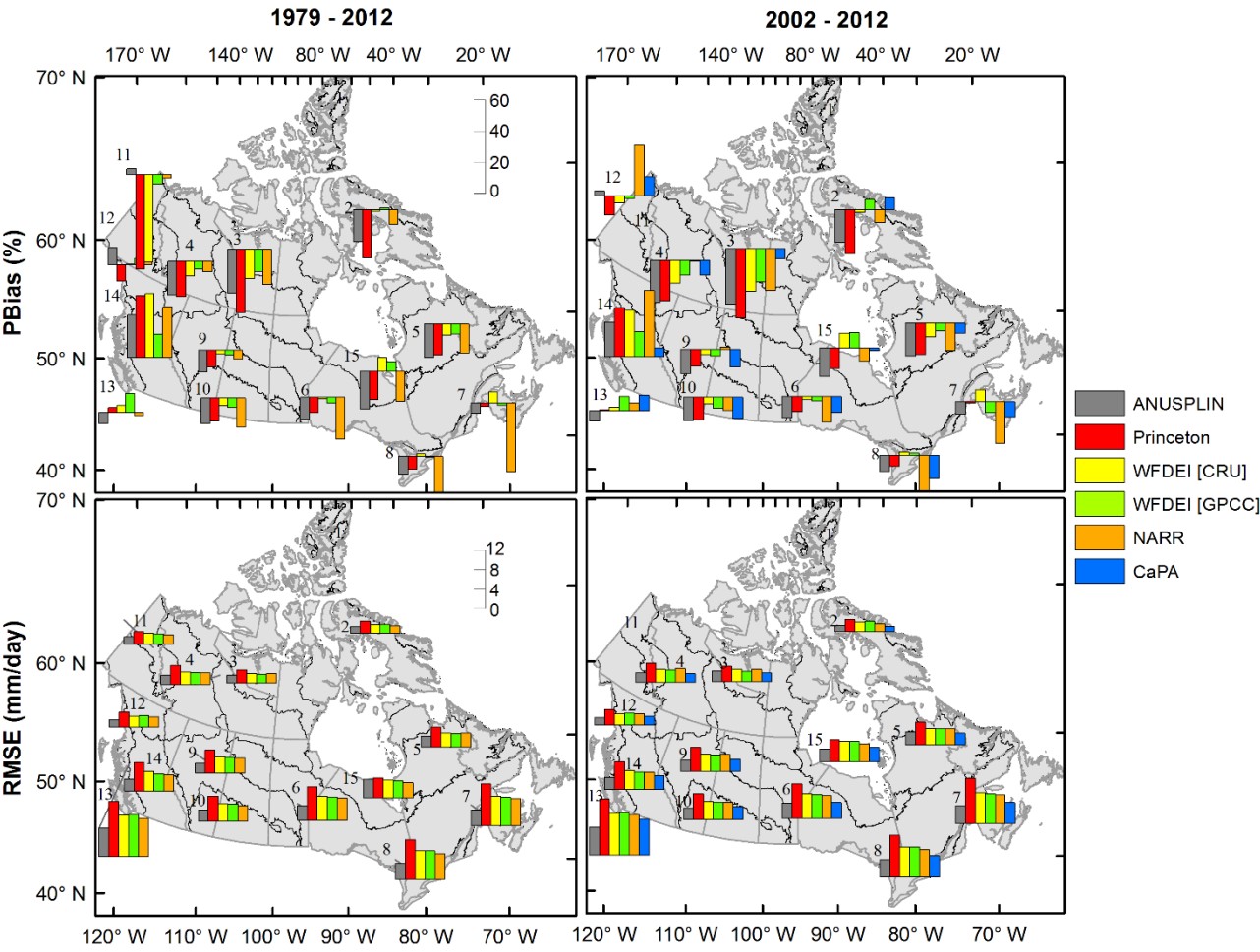

*Figure 9. Bar graphs showing the annual accuracy (PBias) (first row) and magnitude of the errors (RMSE) (second row) of each precipitation dataset for the period of 1979 to 2012 (left panel) and 2002 to 2012 (right panel) in different ecozones. The white bar shows the scale of the bars with number beside it indicating the value of the bar.*