# Peer review of "Inter-comparison of daily precipitation products for large-scale 1 hydro-climatic applications over Canada 2 3 Jefferson S. Wong1\*, Saman Razavi1, Barrie R. Bonsal2, Howard S. Wheater1, & Zilefac E. 4 Asong1 5 6 1 Global Institute for Water Security and School of Environment and Sustainability, University of 7 Saskatchewan, 11 Innovation Blvd, Saskatoon, SK, Canada S7N 3H5 8 2 Environment and Cli"

_Hydrology and Earth System Sciences, 2016_

## Referee Comment (RC1) · Anonymous Referee #1 · 3 Nov 2016

The authors evaluated various gridded precipitation datasets against long-term station data in order to assess accuracy of each datasets. The presence of multiple gridded precipitation datasets available to researchers these days, assessing the accuracy of these datasets (or more appropriately uncertainties in these available datasets) is a very important concern in hydrological research. From the point, this work can be a significant contribution to the hydrologic research for Canada. Despite its importance, I found a number of questions that need to be answered before this one is accepted for publication.

Specific comments:

(1) The authors compare gridded precipitation products against data at individual sta-

tions. A rain gauge data represents only a small area but the gridded data evaluated in this study, especially those based on model products, represent values at much larger area, essentially averages over individual grid boxes. How can we expect that a rain gauge data can represent an average value of hundreds of square kilometers? This must be thoroughly discussed to justify their methodology.

(2) To prepare for evaluations, the authors first interpolated all gridded data into a common grid of 0.5deg resolutions, then they re-interpolated from the grid to the location of individual rain gauges. This data processing includes two spatial interpolations. Because every interpolation step introduces its own errors or uncertainties, the number of interpolation steps must be as small as possible. I wonder why they did not directly interpolate each data set to the rain gauge locations without going through the intermediate grid? This can simply data processing and can reduce interpolation-related uncertainties.

(3) Model products based on RCP scenarios includes the effects of hypothetical emissions pathways implemented in these simulations. How can these model data be compared against the reference data in the same way as other assimilated and/or station-based gridded data? The authors evaluate these data sets for two periods, 1979-2012 and 2002-2012. The CMIP5 experiment that seem relevant to the model data used in this study was designed in such a way that the present-day period simulation based on the realistic GHG concentration for the period from mid-19th century to 2005. Future projections based on specific RCP scenarios starts from 2005 up to 2300 with the initial condition taken at the end of the present-day simulation period. Thus all model data after 2005 are affected by hypothetical emissions pathways. It's pretty unusual discussing "accuracy" of the data generated to project future on the basis of hypothetical GHG concentrations. If the authors are interested in evaluating the model-generated data, the comparison must end in 2005, the end of the present-day period for which the observed external forcing and GHG concentration are implemented. For such periods, there is no need to distinguish runs according to RCP scenarios because the hypo-

thetical emissions pathways are not implemented. My suggestion is to drop model data from the evaluation, or include the model data and limited the evaluation period to 2005 instead of 2012.

(4) The authors provide lengthy descriptions on the details of the data sets used in this study. Much of these discussions are unnecessary because they were developed by other research groups and relevant publications on the details of these data sets are already available. Sections 3.1 and 3.2 can be reduced by referencing suitable publications.

(5) All figures are too busy to read. Need to make them bigger.

---

## Referee Comment (RC2) · Anonymous Referee #2 · 2 Dec 2016

The study examines and compares 8 types of gridded precipitation sources (i.e., 22 precipitation products based on station, reanalysis, and GCM models) over 15 terrestrial ecozones in Canada. I think the results reported by this manuscript can be useful for hydrologists, meteorologists, and potential data users over Canada. In general, the paper is concise and well organized. The results are original and useful for both data developers and end-users, especially for large-scale hydrometeorological applications in Canada. The paper is thus worth to be published after the minor suggestions listed below.

1) The abstract seems too long and needs to be further condensed in the revision. Moreover, the spatiotemporal scales of evaluation (daily and 0.5 deg.) should be de-

[Figure]

noted in the abstract.

2) P4 Line 10-14: In terms of retrieval errors in satellite precipitation, the impact of the snow cover on passive microwave sensors is rather serious over high mountainous regions or high latitude areas, e.g., the Tibetan Plateau (Yong et al., 2015). The authors should address this issue here. Additionally, the Global Precipitation Measurement (GPM; Hou et al. 2014) has been coming and the authors should mention the GPM mission in describing the satellite precipitation estimates. Hou, A. Y., and Coauthors, 2014: The global precipitation measurement mission. Bull. Amer. Meteor. Soc., 95, 701-722. Yong, B., and Coauthors, 2015: Global view of real-time TRMM multisatellite precipitation analysis: Implications for its successor global precipitation measurement mission. Bull. Amer. Meteor. Soc., 96, 283-296.

3) P 17 Line 4-14: Using the approach of Kolmogorov-Smirnov test to evaluate different precipitation products is an interesting way for readers. But here the equation (1) is not clear. I suggest that the authors may carefully re-modified the calculating equation and illustrate the meanings of parameters. If possible, an appendix that introduces the Kolmogorov-Smirnov test might be added at the end of the text. At least, the Eq. (1) should be revised again.

4) P 27 Line 12-14: In the conclusion, please clarify and explain the reasons of the poorest performance of station-based and reanalysis-based products in Atlantic and Pacific regions.

5) Some figures are not very clear and they should be modified or redrawn. For example, there is no whole Canada map (or North American map), no north arrow, no measuring scale in Fig. 1. Figure 2 is OK, but the plots in Fig. 3 and Fig. 4 are too small and not clear for reading. I really hope that these plots could be better displayed in the revised manuscript.

---

## Editor Comment (EC1) · J. Seibert (Editor) · 7 Dec 2016

December 1, 2016

***Paper Summary***

This paper sought to evaluate the performance and reliability of daily gridded precipitation products for Canada - based on seasonality and eco/hydro-zones. The aim of defining specific climatic/hydrological regions and factoring in seasonality was to relay more usability and relatability with the results. The authors identified a need for such a study as few had been done previously which looked at precipitation products for Canada – although they do make reference to a study being conducted previously for "North America".

7 datasets were assessed which fell under 1 of 5 types of precipitation products: *Station-based*, *Station-based model-derived*, *Reanalysis-based multiple-source*, *GCM statistically downscaled* and *GCM-driven RCM dynamically downscaled*. These products were compared against direct precipitation-gauge data from an adjusted and homogenized dataset covering Canada, with the authors acknowledging the scarcity of gauges and lack of quantification of the uncertainty associated with this benchmark dataset. A Kolmogorov-Smirnov test was done to compare the probability distributions of the products and 4 performance measures were carried out: Percentage of Bias, Root-mean square error, Correlation co-efficient and Standard Deviation. Ultimately, the results indicated a strong conclusion was not possible that would name one product superior to all others. Rather, 9 concluding points were presented which cover various regions, seasons and performance measures.

***Main points***

Overall this study does fall under the scope of HESS and has a meaningful aim in assessing the reliability of precipitation products as these same datasets are the ones which feed into hydrological models. This type of work appears to not have been carried out on such a large scale previously, but perhaps setting

out to analyze and summarize 7 datasets, over 15 regions and for all seasons is too grand for a single paper. It is apparent that widespread results exist, as evidenced by the conclusions that the performance of the products depended on both season and eco-zone. An alternative approach to add greater clarity to a project of this size could be to re-structure the format of the paper to present the results based on the zones assessed, perhaps in a tabular format. This would also help users of this study to efficiently compare, contrast and determine the best dataset for their needs (which was an objective of this study). Although the results, discussion and conclusion sections are presented in a convoluted manner, the outcome is still thorough and definitive conclusions are presented. As well, the performance measure methodology is clearly presented and would be easy to reproduce.

The precipitation data section is incredibly unclear. It would first be beneficial to break the section into further components, for example data sources, limitations and treatment. Secondly, the authors have presented a lengthy description on how data was gathered, compiled and corrected, although all of this work was carried out in previous research. What is lacking is a better description toward the end of the section to outline why exactly this reference dataset was selected despite it clearly having major deficiencies.  Three studies are referenced with regards to this dataset being widely used yet no further information is presented. This reference dataset is an integral piece of the analysis, all of the datasets are being compared to it, therefore it is not enough to only state that it "has been recognized".  It would make more sense to outline in detail why it is being used rather than how it came to be as that work has already been done.

This study was done for a large scale and included a number of variables. Textually the results are quite difficult to follow and there is an abundance of figures provided to illustrate these results, but they too are quite dense. A solution would be to either separate, enlarge or regroup the figures to add clarity and

meaning to the results, and by doing so much of the text can be condensed to include key references to the figures without spelling out each result.

***Minor Points***

- Title: the word various does not add any meaning, it can be removed or the count of precipitation products can be used in its place

- Abstract: should list the precipitation products under review, as well, mentions a "systematic analysis framework" but the paper does not read as though any framework has been developed

- Structure and Content: needs reworking.

    o Pg.15 (Line 28) references Section 2.1 which does not exist. Should reference section 3.1 instead.

    o Study area includes a discussion of data collection

    o Introduction should be presented on its own. "*Precipitation measurements and their limitations*" and "*Objectives and Scope"* should not be in the introduction.

    o Most of section 3.2 can be removed and inserted as a summary table as it completely references the outcome of prior studies

- Language: an edit should be conducted to check for grammar and sentence structure. Examples: The results point on Pg.28, Line 15 contains 3 sets of parentheses in a single sentence. The sentence on Pg 7, Line 20 ends with "*along the southern Canada*". Pg.8, Line 4 refers to the province of Alberta as Alberta province.

- References: ample amount of references but this is appropriate given the amount of datasets being analysed. Though several references appear dated, for example the Radar Reflectivity and Surface Rainfall paper likely had several further advances on the topic since 1987

---

## Author Comment (AC1) · 31 Dec 2016

We are grateful to the reviewer for his/her review and comments and suggestions to improve our paper. We have now addressed all of the comments and presented our responses in the attachment.

Please also note the supplement to this comment:
http://www.hydrol-earth-syst-sci-discuss.net/hess-2016-511/hess-2016-511-AC1-supplement.pdf
* * *

---

## Author Comment (AC2) · 31 Dec 2016

**Responses to Reviewer 2 comments on Manuscript HESS-2016-511**

**Title:** Evaluation of various daily precipitation products for large-scale hydro-climatic applications over Canada

Authors: Jefferson Wong et al

Manuscript No: hess-2016-511

The review comments are in regular bold typeface, while all responses are in italics and indented paragraphs.

**Response to Reviewer 2**

The study examines and compares 8 types of gridded precipitation sources (i.e. 22 precipitation products based on station, reanalysis, and GCM models) over 15 terrestrial ecozones in Canada. I think the results reported by this manuscript can be useful for hydrologists, meteorologists, and potential data users over Canada. In general, the paper is concise and well organized. The results are original and useful for both data developers and end-users, especially for large-scale hydrometeorological applications in Canada. The paper is thus worth to be published after the minor suggestions listed below.

We thank the reviewer for reviewing our manuscript and providing his/her valuable comments. We have now addressed all of the comments and presented our responses below.

**Specific comments:**

1. The abstract seems too long and needs to be further condensed in the revision. Moreover, the spatiotemporal scales of evaluation (daily and 0.5 deg.) should be denoted in the abstract.

The length of the abstract will be reduced and the spatiotemporal scales of evaluation will be included in the revised abstract. The following shows the revised abstract, with deleted materials being crossed out by drawing a line through them (and revised sentences being coloured in red):

A number of global and regional gridded climate products based on multiple data sources and models are available that can potentially provide better and more reliable estimates of precipitation for climate and hydrological studies. However, research into the reliability of these products for various regions has been limited and in many cases non-existent. This study identifies several gridded precipitation products and over Canada and develops a systematic analysis framework to assess the characteristics of errors associated with the different datasets, using the best available adjusted precipitation-gauge data as a benchmark over the period 1979 to 2012. The framework quantifies the spatial and temporal variability of the errors over 15 terrestrial ecozones in Canada for different seasons over the period 1979 to 2012 at 0.5° and daily spatiotemporal resolution at the daily time scale. Results showed that most of the products were relatively skillful in central Canada. However, they tended to overestimate precipitation amounts on the west coast and underestimate on the east and especially in northern Canada (above  $60^{\circ}$  N). but tended to underestimate precipitation amounts on the east coast and overestimate on the west. The global product by WATCH Forcing Data ERA-Interim (WFDEI) augmented by Global Precipitation Climatology Centre (GPCC) data (WFDEI [GPCC]) performed best with respect to different metrics. The Canadian Precipitation Analysis (CaPA) product of Meteorological Service of Canada, performed comparably with WFDEI [GPCC], however it only provides data from 2002. All the products performed best in summer, followed by autumn, spring, and winter in order of decreasing quality. Due to the sparse observational network, northern Canada (above  $60^{\circ}N$ ) was most difficult to assess with the majority of products tending to significantly underestimate total precipitation. Results from this study can be used as a guidance for potential users regarding the performance of different precipitation products for a range of geographical regions and time periods.

2. P4 Line 10-14: In terms of retrieval errors in satellite precipitation, the impact of the snow cover on passive microwave sensors is rather serious over high mountainous regions or high latitude areas, e.g. the Tibetan Plateau (Yong et al., 2015). The authors should address this issue here. Additionally, the Global Precipitation Measurement (GPM; Hou et al. 2014) has been coming and the authors should mention the GPM mission in describing the satellite precipitation estimates. Hou, A. Y., and Coauthors, 2014: The global precipitation measurement mission. Bull. Amer. Meteor. Soc., 95, 701-722. Yong, B., and Coauthors, 2015: Global view of real-time TRMM multisatellite precipitation analysis: Implications for its successor global precipitation measurement mission. Bull. Amer. Meteor. Soc., 96, 283-296.

The impact of snow cover on passive microwave sensors will be addressed and the GPM mission will be mentioned in the revised manuscript. Accordingly, the corresponding references will also be added. The following shows the revised discussion of satellite-based estimates in the original manuscript [P4:L7-14], with additional sentences being coloured in red:

Development of satellite-based precipitation estimates has provided coverage over vast gauged/ungauged regions with continuous observations regardless of time of day, terrain, and weather condition of the ground (Gebregiorgis and Hossain, 2015). The recently launched Global Precipitation Measurement (GPM) Core Observatory has further opened up new opportunities for observing worldwide precipitation from space (Hou et al., 2014). However, satellite-based estimates also contain inaccuracies resulting primarily from temporal sampling errors due to infrequent satellite visits to a particular location, instrumental errors due to calibration and measurement noise, and algorithm errors related to approximations to the cloud physics used (Nijssen and Lettenmaier, 2004;Gebremichael et al., 2005). In particular, the passive microwave overpasses were shown to be unreliable over regions with snow cover and complex terrain such as the Tibetan Plateau (Yong et al., 2015).

3. P17 Line 10-14: Using the approach of Kolmogorov-Smirnov test to evaluate different precipitation products is an interesting way for readers. But here the equation (1) is not clear. I suggest that the authors may carefully re-modified the calculating equation and illustrate the meanings of parameters. If possible, an appendix that introduces the Kolmogorov-Smirnov test might be added at the end of the text. At least, the Eq. (1) should be revised again.

We will address this comment by providing better explanation of the calculation and revising the wordings in the equation for better clarity in the revised manuscript. The following shows the revised Sections 4.1, with deleted materials being crossed out by drawing a line through them and revised sentences being coloured in red:

A two-sample non-parametric Kolmogorov-Smirnov (K-S) test compared was used to compare the cumulative distribution functions (CDFs) for of each type of gridded precipitation product and the ground observations. at 5 % significance level ( $\alpha =$ 0.05) to support the The null hypothesis (H0) for this test is that the two datasets came from same population. Monthly total precipitation data were used and aggregated for each season because the existence of numerous zero values in the daily precipitation data might reduce the statistical identification of significant differences to support the null hypothesis. The K-S test was repeated independently for all precipitation-gauge stations at 5 % significance level ( $\alpha = 0.05$ ). and a measure of reliability (in percent) was calculated to show how reliable each type of precipitation products was among all the precipitation-gauge stations, as shown by Eq. (1) A measure of reliability (in percent) was calculated based on counting the numbers of stations that do not reject the null hypothesis (any p values greater than 0.05) over the total numbers of stations (145 and 137 stations in long-term and short-term comparison respectively), which is shown by Eq. (1).

$$\frac{\text{no of station that support } H_{0}}{\text{total no of precipitation gauge station}} \cdot 100 \tag{1}$$

% of reliability =
$$\frac{numbers of stations that support H_0}{total numbers of precipitation gauge stations} \cdot 100$$
 (1)

We appreciate the suggestion on having an appendix to introduce the Kolmogorov-Smirnov (K-S) test but we decide not to do so due to the following reasons: (1) K-S test is one of the most commonly-used statistical tests and its basic theory, assumptions, and calculation is easily found in any statistical handbooks; (2) we only applied the standard two-sample non-parametric K-S test in our study without any modifications in its assumptions or calculation; and (3) given the length of our manuscript, we prefer saving the space for better explanation or clarification in other parts of the manuscript (if necessary).

4. P27 Line 12-14: In the conclusion, please clarify and explain the reasons of the poorest performance of station-based and reanalysis-based products in Atlantic and Pacific regions.

We think that this statement in the Conclusion [P27:L12-14] of the original manuscript will cause some confusions and we decide to drop it from the conclusion and address the reasons of the poor performance in the Results Section (Section 5.2) [P22:L23] in the revised manuscript, which is shown as follows:

The resulting values of the RMSE metric in Regions 7 (Atlantic Maritime) and 13 (Pacific Maritime) tended to be larger than that of other areas. However, the other metrics such as correlation coefficient and PBias showed better performance in these regions. This suggests that higher RMSE values can be mainly attributed to the fact that precipitation amounts are higher in the maritime regions.

5. Some figures are not very clear and they should be modified or redrawn. For example, there is no whole Canada map (or North American map), no north arrow, no measuring scale in Fig. 1. Figure 2 is OK, but the plots in Fig. 3 and Fig. 4 are too small and not clear for reading. I really hope that these plots could be better displayed in the revised manuscript.

We agree that some of the figures are not very clear as it is also commented by Reviewer 1. We will enlarge the figures as much as possible and provide the missing map information in Figure 1 in the revised manuscript. In response to comment 3 of Reviewer 1, we decide to limit the evaluation period to 2005 instead of 2012 for the climate model products. Accordingly, Figures 2, 3, and 4 in the original manuscript will be reproduced to reflect the change. In short, the evaluation for the climate model products from the period of 1979 to 2005 will be shown separately from that of station-based and reanalysis-based products. Thus, Figures 3 and 4 will only show the distributions of pvalue of the K-S test for the station-based and reanalysis-based products and a new Figure 5 will be created to show the distributions of p-value of the K-S test for climate model products in the revised manuscript. The numbering of Figures 5 to 8 will also be changed accordingly. Note that all the figures in the supplementary materials will also be subject to the same changes as aforementioned but will not be shown here. The revised figures are shown as follows:

---

## Author Comment (AC3) · 31 Dec 2016

**Responses to Interactive comments on Manuscript HESS-2016-511**

**Title:** Evaluation of various daily precipitation products for large-scale hydro-climatic applications over Canada

**Authors:** Jefferson Wong et al

**Manuscript No:** hess-2016-511

The review comments are in regular bold typeface, while all responses are in italics and indented paragraphs.

**Response to Reviews by MSc student**

**Paper Summary**
**This paper sought to evaluate the performance and reliability of daily gridded precipitation products for Canada – based on seasonality and eco/hydro-zones. The aim of defining specific climatic/hydrological regions and factoring in seasonality was to relay more usability and relatability with the results. The authors identified a need for such study as few had been done previously which looked at precipitation products for Canada – although they do make reference to a study being conducted previously for "North America".**

**7 datasets were assessed which fell under 1 of 5 types of precipitation products: station-based, station-based model-derived, Reanalysis-based multiple-source, GCM statistically downscaled and GCM-driven RCM dynamically downscaled. These products were compared against direct precipitation-gauge data from an adjusted and homogenized dataset covering Canada, with the authors acknowledging the scarcity of gauges and lack of quantification of the uncertainty associated with this benchmark dataset.**

**A Kolmogorov-Smirnov test was done to compare the probability distributions of the products and 4 performance measures were carried out: Percentage of Bias, Root-mean-square-error, Correlation coefficient and Standard Deviation. Ultimately, the results indicated a strong conclusion was not possible that would name one product superior to all others. Rather, 9 concluding points were presented which cover various regions, seasons and performance measures.**

> *We thank the reviewer for reviewing our manuscript and providing a very nice summary of our work. We have now addressed all of the comments and presented our responses below, with deleted materials being crossed out by drawing a line through them and revised sentences being coloured in red.*

**Main points:**

1. **Overall this study does fall under the scope of HESS and has a meaningful aim in assessing the reliability of precipitation products as these same datasets are the ones which feed into hydrological models. This type of work appears to no have been carried out on such a large scale previously, but perhaps setting out to analyze and summarize 7 datasets, over 15 regions and for all seasons is too grand for a single paper. It is apparent that widespread results exist, as evidenced by the conclusions that the performance of the products depended on both season and eco-zone. An alternative approach to add greater clarity to a project of this size could be to re-structure the format of the paper to present the results based on the zones assessed, perhaps in a tabular format. This would also help users of this study to efficiently compare, contrast and determine the best dataset for their needs (which was an objective of this study). Although the results, discussion and conclusion sections are presented in a convoluted manner, the outcome is still thorough and definitive conclusions are presented. As well, the performance measure methodology is clearly presented and would be easy to reproduce.**

   *We appreciate the value of the reviewer's suggestion on the format of the presentation of results. We agree that presenting the results based on ecozones in a tabular format would be very efficient to compare and contrast only when several datasets (e.g. three to four) over a few regions (e.g. up to five) are involved in the analysis (i.e. up to 20 numbers in a table). However, when more datasets and more regions are involved, such as in our case (six datasets over 15 ecozones), efficiency might be significantly reduced when going through a tabular table with 90 numbers. We have already thought about different ways to present and summarize our results (e.g. tabular table, Taylor diagram, line graph, box and whisker plot) and identified portrait diagram (Figures 5 and 6 in the original manuscript), which is widely used in climate models comparison studies (e.g. Pincus et al., 2008;Sillmann et al., 2013), is the most suitable way to show the results which can highly condense information in one diagram.*

2. **The precipitation data section is incredibly unclear. It would first be beneficial to break the section into further components, for example data sources, limitations and treatment. Secondly, the authors have presented a lengthy description on how data was gathered, complied and corrected, although all of this work was carried out in previous research.**

   *We agree that we have a lengthy data description section as it is also commented by Reviewers 1. The details in Sections 3.1 and 3.2 will be greatly reduced in the revised manuscript. In short, the spatial and temporal resolutions of each product, their compositions, and examples of their applications will be remained and other details will be deleted. The following shows the revised Sections 3.1 and 3.2:*

**3.1 Precipitation-gauge station data**

*Climate data collection is coordinated by the Federal government of Canada. Agriculture and Agri-Food Canada maintains a few stations nationally especially in  province of Alberta. Also, most hydro-power companies collect their own data. However, their data are not made available to the public but are sent to Environment and Climate Change Canada for archiving prior to release. In other words, the National Climate Data Archive of Environment Canada provide the basis for all the available climate data. Based on the National Climate Data Archive of Environment Canada, there are a total of 1499 precipitation-gauge stations (as in 2012) across Canada. However, due to the addition and subtraction of climate stations over the past few decades, the number of stations with available precipitation data for specified time intervals varies greatly. For instance, the numbers of precipitation-gauge stations that were active in any given years over the period of 1961 to 2003 ranged from 2000 to 3000 (see Hutchinson et al. (2009) Figs 1 and 2 for details). The issue with these data is they are subject to various errors, among which the errors due undercatch are quite significant in Canada (Mekis and Hogg, 1999). In order to account for various measurement issues, Mekis and Vincent (2011) provided adjusted daily rainfall and snowfall data for 464 stations over Canada that were based on the Adjusted Precipitation for Canada dataset (Mekis and Hogg, 1999). The data extend back to 1895 for a few long-term stations and run through 2014. For these data, daily rainfall gauge and snowfall ruler data were extracted from the National Climate Data Archive of Environment Canada and adjustments of rain and snow were done separately (Devine and Mekis, 2008) (Mekis and Brown, 2010).  Observations from nearby stations were sometimes combined to create longer time series  (Vincent and Mekis, 2009). As a result of adjustments, total rainfall amounts were concluded to be 5 to 10 % higher in southern Canada and more than 20 % in the Canadian Arctic than the original observations. The effect of the adjustments on snowfall were larger and more variable throughout the country. Despite the lack of a measure of associated uncertainty, this adjusted precipitation-gauge station dataset has been recognized and widely used for different analyses (e.g. Nalley et al., 2012;Shook and Pomeroy, 2012;Wan et al., 2013). Therefore, this dataset was used in this study as the reference to represent the best available precipitation measurement and as the benchmark for all gridded precipitation product comparisons.*

*3.2  Gridded precipitation products*

*Seven precipitation datasets were assessed. Table 1 provides a concise summary of these datasets, including their full names, and original spatial and temporal resolutions for the versions used. These particular datasets were chosen based on the following criteria: (1) a complete coverage of Canada; (2) minimum of daily temporal and 0.5° (~50 km) spatial resolutions; (3) sufficient lengths of data (>30 years) for long-term study and cover recent years up to 2012; and (4) representation of a range of sources/methodologies (e.g. station based, remote sensing, model, blended products). Note that other commonly used datasets including the monthly Canadian Gridded temperature and precipitation (CANGRD) dataset (Zhang et al., 2000) and the coarser resolution Japan Meteorological Agency 55-year Reanalysis (JRA-55) (Onogi et al., 2007;Kobayashi et al., 2015) and the Modern-Era Retrospective Analysis for Research and Applications (MERRA) (Rienecker et al., 2011) products were excluded as they do not meet criteria # 2 above.*

*3.2.1  Station-based product – ANUSPLIN*

*With the application of the Australian National University Spline (ANUSPLIN) model (Hutchinson, 1995;Hutchinson, 2004), Hutchinson et al. (2009)  developed a climate dataset of daily precipitation and daily minimum and maximum air temperature over Canada at a spatial resolution of 300 arc-second of latitude and longitude (0.0833° or ~10 km) for the period of 1961 to 2003, using observed stations (from 2000 to 3000 in any given years over the period) recorded in the National Canadian Climate Data Archives of Environment Canada. However, to retain a better spatial coverage, no adjustments were done on the archive station data before the generation of the product. The dataset was generated to model the complex spatial patterns by using tri-variate thin-plate smoothing splines method that incorporated spatially continuous functions of latitude, longitude, and elevation. Hopkinson et al. (2011) subsequently extended this original dataset to include the period of 1950 to 2011. This ANUSPLIN product for Canada (hereafter the ANUSPLIN)*  *has further been updated to 2013 and has recently been used as the basis of 'observed' data for evaluating different climate datasets (e.g. Eum et al., 2012) and for assessing the effects of different climate products in hydrological applications (e.g. Eum et al., 2014;Bonsal et al., 2013;Shrestha et al., 2012a).*

*3.2.2  Station-based model-derived product – CaPA*

*Initiated in November 2003 through collaborations within the Meteorological Service of Canada, the Canadian Precipitation Analysis (CaPA) was developed to produce a dataset*

of 6-hourly precipitation accumulation over North America in real-time at a spatial resolution of 15 km from 2002 onwards (Mahfouf et al., 2007). The dataset was generated based on an optimum interpolation technique (Daley, 1993), which required a background field and a specification of error statistics between the observations and the background field (e.g. Bhargava and Danard, 1994;Garand and Grassotti, 1995). For Canada, the short-term precipitation forecasts from the Canadian Meteorological Centre (CMC)'s regional model, the Global Environmental Multiscale (GEM) (Cote et al., 1998a;1998b), were used as the background field with the rain-gauge measurements from the observational network as the observations. ~~The analysis was created by simple kriging to interpolate the differences between the transformed data of GEM and stations, which was then re-transformed and applied back to GEM. The quality of rain-gauge stations was controlled by cross-checking with the neighbouring stations and by comparing with the radar-derived precipitation. The accuracy of the product was assessed by generating an analysis error that represented the amount of additional information gained from the multiple observations with regard to the background field.~~ CaPA has become operational at the CMC in April 2011, with updates to the statistical interpolation method (Lespinas et al., 2015), increase of spatial resolution to 10 km and the assimilation of Quantitative Precipitation Estimates from the Canadian Weather Radar Network as an additional source of observations (Fortin et al., 2015b). With its continuous improvement and different configurations, CaPA has been employed in Canada for various environmental prediction applications (e.g. Eum et al., 2014;Fortin et al., 2015a;Pietroniro et al., 2007;Carrera et al., 2015). However, the study period of these applications only extended back to 2002.

*3.2.3 Reanalysis-based multiple-source products – Princeton, WFDEI, and NARR*

**Princeton**

The Terrestrial Hydrology Research Group at the Princeton University initially developed a dataset of 3-hourly near-surface meteorology with global coverage at a $1.0°$ spatial resolution (~120 km) from 1948 to 2000 for driving land surface models and other terrestrial systems (Sheffield et al., 2006). The global dataset at the Princeton University (called hereafter the "Princeton") was constructed based on the National Centers for Environmental Prediction-National Center for Atmospheric Research (NCEP-NCAR) reanalysis ($2.0°$ and 6-hourly) (Kalnay et al., 1996;Kistler et al., 2001), combining with a suite of global observation-based data including the Climatic Research Unit (CRU) monthly climate variables (New et al., 2000, 1999), the Global Precipitation Climatology Project (GPCP) daily precipitation (Huffman et al., 2001), the Tropical Rainfall Measuring Mission (TRMM) 3-hourly precipitation (Huffman et al., 2002), and the NASA Langley Research Center monthly surface radiation budget (Gupta et al., 1999).

*from TRMM data, and the sophistication of the correction methods (a correction to the wet-day statistics (Sheffield et al., 2004), and monthly bias corrections to match those of the CRU data (Adam and Lettenmaier, 2003)). The Princeton dataset has been evaluated against the Second Global Soil Wetness Project (GSWP-2) product (Zhao and Dirmeyer, 2003).* *With the inclusion of new temperature and precipitation data (e.g. Willmott et al., 2001), Princeton has been updated and is currently available at $1.0°$ (plus $0.5°$ and $0.25°$), 3-hourly (plus daily and monthly) resolution globally for 1948 to 2008. Experimental updates including a 1901-2012 version at $1.0°$ (plus $0.5°$), 3-hourly (plus daily and monthly) resolution are also available. Studies employing Princeton to study different hydrological aspects have been carried out over different parts of Canada (e.g. Kang et al., 2014;Su et al., 2013;Wang et al., 2013;Wang et al., 2014).*

**WFDEI**

*To simulate the terrestrial water cycle using different land surface models and general hydrological models, the European Union Water and Global Change (WATCH) Forcing Data (WFD) were created to provide datasets of sub-daily (3-hourly or 6-hourly) and daily meteorological data with global coverage at a $0.5°$ spatial resolution (~50 km) from 1901 to 2001 (Weedon et al., 2011). Similar to the composition of the Princeton dataset, the WFD were derived from the 40-year European Centre for Medium-Range Weather Forecasts (ECMWF) Re-Analysis (ERA-40) ($1.0°$ and 3-hourly) (Uppala et al., 2005) and combined with the CRU monthly variables and the Global Precipitation Climatology Centre (GPCC) monthly data (Rudolf and Schneider, 2005;Schneider et al., 2008;Fuchs, 2009).* *The generation of the WFD for 1958 to 2001, which was based on the ERA-40, followed the procedures developed by Ngo-Duc et al. (2005) and Sheffield et al. (2006) whereas the dataset for 1901 to 1957 was generated by using the reordered ERA-40 a year at a time. With respect to precipitation, the creation of the data (Weedon et al., 2010) involved spatially downscaling using the CRU data, sequential elevation correction, wet-day correction, monthly precipitation bias correction to match the GPCC data, and adjustment for gauge undercatch (Adam and Lettenmaier, 2003), however no corrections were made for orography effect (Adam et al., 2006). The same monthly bias corrections were also done using the CRU precipitation totals, resulting in two sets of precipitation data. The WFD were assessed by the FLUXNET data for selected years at seven sites (Araujo et al., 2002;Persson et al., 2000;Suni et al., 2003;Meyers and Hollinger, 2004;Grunwald and Bernhofer, 2007;Urbanski et al., 2007;Gockede et al., 2008).* *The WATCH Forcing Data methodology applied to ERA-Interim (WFDEI) dataset has further been generated covering the period of 1979 to 2012 (Weedon et al., 2014). The WFDEI used the same methodology as the WFD, but based on the ERA-Interim (Dee et al., 2011) with higher spatial resolution ($0.7°$)* *, better data assimilation technique, updated monthly observation-based data, more extensive incorporation of observations, and correction of the most extreme cases of inappropriate precipitation phase.* *As for the WFD, the WFDEI*

had two sets of rainfall and snowfall data generated by using either CRU or GPCC precipitation totals (hereafter the WFDEI [CRU] and WFDEI [GPCC] respectively). To date, specific studies using the WFDEI related to Canada has been limited to the studies of permafrost in the Arctic regions (e.g. Chadburn et al., 2015;Park et al., 2015;Park et al., 2016) but the WFDEI could be a potential source in other environmental applications in Canada.

***NARR***

*Concerning the spatial and temporal water availability in the atmosphere, the North American Regional Reanalysis (NARR) was developed to provide datasets of 3-hourly meteorological data for the North America domain at a spatial resolution of 32 km (~0.3°) covering the period of 1979 to 2003 as the retrospective system and is being continued in near real-time (currently up to 2015) as the Regional Climate Data Assimilation System (R-CDAS) (Mesinger et al., 2006). The components in generating NARR included the NCEP-DOE reanalysis (Kanamitsu et al., 2002), the NCEP regional Eta Model (Mesinger et al., 1988;Black, 1988) and*  *the Noah land-surface model (Mitchell et al., 2004;Ek et al., 2003), and the use of numerous additional data sources (see Mesinger et al., 2006 Table 2).* ~~The use of NCEP-DOE reanalysis was a major improvement upon the earlier NCEP-NCAR reanalysis in both resolution and accuracy to provide lateral boundary conditions. Regarding precipitation assimilation scheme, the NARR adjusted the accumulated convective and grid-scale precipitation, assimilated the precipitation observations as latent heating profiles based on the differences between the modelled and observed precipitation (Lin et al., 1999), and disaggregated into hourly resolution using different sources over lands and oceans. For the period from 1979 to 2003 when NARR was run as the retrospective system, precipitation analyses over the continental United States (CONUS), Mexico, and Canada were derived solely from a gridded analysis of 24-hour rain-gauge measurements. For the period from 2004 onwards, NARR was generated in near-real time by the R-CDAS, which was identical to the retrospective NARR except for changes in input sources and their processing because of the real-time production constraints. One of the major differences was the use of radar-dominated precipitation analyses derived from the National Land Data Assimilation System (NLDAS) (Mitchell et al., 2004) over CONUS to disaggregate the 24-hour rain-gauge analysis to hourly precipitation whereas no assimilation was done over Canada due to the paucity of rain-gauge observations.* On the basis of hydrological modelling in Canada, Choi et al. (2009) found that NARR provided reliable climate inputs for northern Manitoba while Woo and Thorne (2006) concluded that NARR had a cold bias resulting in later snowmelt peaks in subarctic Canada. In addition, Eum et al. (2012) identified a structural break point in the NARR dataset over the Athabasca River basin.*

*3.2.4 GCM statistically downscaled products – PCIC*

*The Pacific Climate Impacts Consortium (PCIC), which is a regional climate service centre at the University of Victoria, British Columbia, has offered datasets of statistically downscaled daily precipitation and daily minimum and maximum air temperature under three different Representative Concentration Pathways (RCPs) scenarios (RCP 2.6, RCP 4.5, and RCP 8.5) (Meinshausen et al., 2011) over Canada at a spatial resolution of 300 arc-second (0.833° or ~10 km) for the historical and projected period of 1950 to 2100 (Pacific Climate Impacts Consortium; University of Victoria, Jan 2014). These downscaled datasets were a composite of 12 GCM projections from the Coupled Model Inter-comparison Project Phase 5 (CMIP5) (Taylor et al., 2012) and the ANUSPLIN dataset. The historical 1950 to 2005 period of the ANUSPLIN was used to drive the GCMs and the statistical properties and spatial patterns of the downscaled outputs tended to resemble those of the ANUSPLIN. However, the timing of natural climate variability (e.g. El Niño-Southern Oscillation) in the observational record were not considered since GCMs were solved as a 'boundary value problem'. Two different downscaling methods were used to downscale to a finer resolution. The first one was Bias Correction Spatial Disaggregation (BCSD) (Wood et al., 2004) following Maurer and Hidalgo (2008) and the second was Bias Correction Constructed Analogues (BCCA) with Quantile mapping reordering (BCCAQ) which was a post-processed version of BCCA (Maurer et al., 2010). In general, the most important distinction between the two methods was BCCAQ obtained spatial information from a linear combination of historical analogues for daily values and retained the daily sequencing of weather events from the coarse resolution, while BCSD only used monthly averages to reconstruct daily patterns by randomly resampling a historic month and scaling its daily values to match the monthly projected values. The ensemble of the PCIC dataset has currently been used in studying the hydrological impacts of climate change on river basins mainly in British Columbia (e.g. Shrestha et al., 2011;Shrestha et al., 2012b;Schnorbus et al., 2014) and Alberta (e.g. Kienzle et al., 2012;Forbes et al., 2011) in Canada. In this study, only four GCMs with two respective statistically downscaling methods under RCP 4.5 and 8.5 were chosen for comparison (see Table 2 for details). The choice of selecting the four GCMs under RCP 4.5 and 8.5 only in the PCIC dataset was to match those GCMs available in the NA-CORDEX dataset (see next section for details).*

*3.2.5 GCM-driven RCM dynamically downscaled products – NA-CORDEX*

*Sponsored by the World Climate Research Programme (WCRP), the COordinated Regional climate Downscaling EXperiment (CORDEX) over North America domain (NA-CORDEX) was launched to provide dynamically downscaled datasets of 3-hourly or daily meteorological data over most of North America (below 80° N) at two spatial resolutions of 0.22° and 0.44° (or 25 and 50 km) under two different RCPs (RCP 4.5, and RCP 8.5) for the historical and projected period of 1950 to 2100 (Giorgi et al., 2009). Within the NA-CORDEX framework, a matrix of six GCMs from the CMIP5 driving six different RCMs was selected to compare the performance of RCMs and characterize the uncertainties*

*underlying regional climate change projections and thus provided climate scenarios for* *further impact and adaption studies.* On top of the knowledge and experience gained from the North American Regional Climate Change Assessment Program (NARCCAP) (Mearns et al., 2012), *a matrix of six GCMs from the CMIP5 driving six different RCMs was selected to compare the performance of RCMs and characterize the uncertainties underlying regional climate change projections and thus provided climate scenarios for further impact and adaption studies.* *the selection of GCM-RCM matrix of simulations, with higher spatial resolution and greater sampling of uncertainty, was based on model climate sensitivity and quality of boundary conditions. In addition, to determine the large variations in future climate due to internal variability of the GCMs on downscaled outputs, samples among multiple realizations of GCM simulations were used to drive the RCMs. The performance of participating RCMs in reproducing historical and projected climate was then assessed by comparing the ERA-Interim-driven RCM simulations.* Current studies using NA-CORDEX datasets were mainly focused on evaluating the model performance of different GCM-driven RCM simulations over North America (e.g. Lucas-Picher et al., 2013;Martynov et al., 2013;Separovic et al., 2013) but the NA-CORDEX dataset could also be a potential source in hydro-climatic studies in Canada. In this study, only two GCMs with three RCMs were chosen for comparison due to the availability of the NA-CORDEX dataset (see Table 3 for details).*

3. **What is lacking is a better description toward the end of the section to outline why exactly this reference dataset was selected despite it clearly having major deficiencies. Three studies are referenced with regards to this dataset being widely used yet no further information is presented. This reference dataset is an integral piece of the analysis, all of the datasets are being compared to it, therefore it is not enough to only state that it "has been recognized". It would make more sense to outline in detail why it is being used rather than how it came to be as that work has already been done.**

> *We will further explain and justify the reasons of using Mekis and Vincent (2011) as our reference in the revised manuscript, which is shown as follows:*

>> *Despite the lack of a measure of associated uncertainty, this adjusted precipitation-gauge station dataset has been recognized and widely used for different analyses (e.g. Nalley et al., 2012;Shook and Pomeroy, 2012;Wan et al., 2013). Since there are no readily reliable daily gridded precipitation data for Canada as viable alternatives,* *Therefore,* *this dataset was* *therefore* *used in this study as the reference to represent the best available precipitation measurement and as the benchmark for all gridded precipitation product comparisons.*

4. **This study was done for a large scale and included a number of variables. Textually the results are quite difficult to follow and there is an abundance of figures provided to illustrate these results, but they too are quite dense. A solution would be to either separate, enlarge or regroup the figures to add clarity and meaning to the results, and**

**by doing so much of the text can be condensed to include key references to the figures without spelling out each result.**

> *We agree that some of the figures are too dense as it is also commented by both Reviewers 1 and 2. However, we believe that Figures 1, 5 to 8 are clear enough to show the messages and therefore we will only enlarge the figures as much as possible in the revised manuscript. In response to comment 3 of Reviewer 1, we decide to limit the evaluation period to 2005 instead of 2012 for the climate model products. Accordingly, Figures 2, 3, and 4 in the original manuscript will be reproduced to reflect the change. In short, the evaluation for the climate model products from the period of 1979 to 2005 will be shown separately from that of station-based and reanalysis-based products. Thus, Figures 3 and 4 will only show the distributions of p-value of the K-S test for the station-based and reanalysis-based products and a new Figure 5 will be created to show the distributions of p-value of the K-S test for climate model products in the revised manuscript. The numbering of Figures 5 to 8 will also be changed accordingly. Note that all the figures in the supplementary materials will also be subject to the same changes as aforementioned but will not be shown here. The revised figures are shown as follows:*

[Figure]

Figure 1. The percentage of reliability, calculated by the Eq. (1), of each precipitation dataset in four seasons for the period of 1979 to 2012 (left panel) and 2002 to 2012 (right panel) across Canada. The higher the percentage, the more reliable the precipitation dataset. Different colours represent different precipitation products, with magenta representing the whole PCIC datasets and cyan representing the whole NA-CORDEX datasets. The full names of the precipitation products are provided in Tables 1, 2, and 3.

[Figure]

Figure 2. Distributions of p-value of the K-S test in the 15 ecozones in four seasons for the period of 1979 to 2012 (long-term comparison without CaPA). Note that the numbers of precipitation-gauge stations in each ecozone are different (see Table 4). Each hollow circle represents one p-value of the K-S test conducted at one precipitation-gauge station, with no stations in Region 1 (R1). The p-values of Regions 6 to 9, and 13 to 14 (R6-R9, and R13-R14), which have more than or equal to 10 stations, were shown in box-whisker plots with bottom, band (black thick line) and top of the box indicating the 25th, 50th (median), and 75th percentiles, respectively.

[Figure]

Figure 3. Distributions of p-value of the K-S test in the 15 ecozones in four seasons for the period of 2002 to 2012 (short-term comparison with the inclusion of CaPA). Note that the numbers of precipitation-gauge stations in each ecozone are different (see Table 4). Each hollow circle represents one p-value of the K-S test conducted at one precipitation-gauge station. The percentage of missing values in precipitation-gauge station in Region 11 (R11) exceeded 10% and thus no K-S test was conducted. The p-values of Regions 6, 8 to 9, and 13 to 14 (R6, R8-R9, and R13-R14), which have more than or equal to 10 stations, were shown in box-whisker plots with bottom, band (black thick line) and top of the box indicating the 25th, 50th (median), and 75th percentiles, respectively.

**1979 - 2005**

[Figure]

Figure 4. Distributions of p-value of the K-S test in the 15 ecozones in four seasons for the period of 1979 to 2005 (long-term comparison of PCIC and NA-CORDEX). Note that the numbers of precipitation-gauge stations in each ecozone are different (see Table 4). Each hollow circle represents one p-value of the K-S test conducted at one precipitation-gauge station, with no stations in Region 1 (R1). The p-values of Regions 6 to 9, and 13 to 14 (R6-R9, and R13-R14), which have more than or equal to 10 stations, were shown in box-whisker plots with bottom, band (black thick line) and top of the box indicating the 25th, 50th (median), and 75th percentiles, respectively.

**Minor Points**.

5.  **Title: the word various does not add any meaning. It can be removed or the count of precipitation products can be used in its place.**

    *We agree that the word various does not add much meaning in the tile and we decide to remove the word in the revised manuscript. Also, in response to comment 2 of Reviewer 1, we will change the title from "Evaluation" to "Inter-comparison" to better reflect our aim of the study. The title in the revised manuscript will become:*

    > *Inter-comparison of daily precipitation products for large-scale hydro-climatic applications over Canada*

6.  **Abstract: should list the precipitation products under review, as well, mentions a "systematic analysis framework" but the paper does not read as though any framework has been developed.**

    *We fully understand that it is essential to list the precipitation products under review in the abstract. However, given the numbers of precipitation products we analyzed and the length of the full names of the products, listing the products in the abstract takes so much room which then limit the messages we can deliver from our study. Therefore, we prefer saving the space for telling the main findings of our study which are more important to the readers and decide not to add the list of the products in the revised abstract. We will delete "systematic analysis framework" and reduce the length of the abstract when responding to comment 1 of Reviewer 2, which is shown as follows:*

    > *A number of global and regional gridded climate products based on multiple data sources and models are available that can potentially provide*  *more reliable estimates of precipitation for climate and hydrological studies. However, research into the reliability of these products for various regions has been limited and in many cases non-existent. This study identifies several gridded precipitation products* and  *quantifies the spatial and temporal variability of the errors over 15 terrestrial ecozones in Canada for different seasons* over the period 1979 to 2012 at 0.5$^{\circ}$ and daily spatiotemporal resolution *. Results showed that most of the products were relatively skillful in central Canada.* However, they tended to overestimate precipitation amounts on the west coast and underestimate on the east and especially in northern Canada (above 60$^{\circ}$ N). *. The global product by WATCH Forcing Data ERA-Interim (WFDEI) augmented by Global Precipitation Climatology Centre (GPCC) data (WFDEI [GPCC]) performed best with respect to different metrics. The Canadian Precipitation Analysis (CaPA) product of Meteorological Service of Canada,*

*performed comparably with WFDEI [GPCC], however it only provides data from 2002. All the products performed best in summer, followed by autumn, spring, and winter in order of decreasing quality.  Results from this study can be used as a guidance for potential users regarding the performance of different precipitation products for a range of geographical regions and time periods.*

7. **Structure and Content: needs reworking.**

- **P15:L28: references Section 2.1 which does not exist. Should reference section 3.1 instead.**
  *Thank you for spotting out this mistake. We will correct the referencing to Section 3.1 in the revised manuscript, which is shown as follows:*

  > *To identify the most consistent gridded dataset corresponding to different seasons and regions across Canada, comparisons of each gridded product with direct precipitation-gauge station data from the Canadian adjusted and homogenized precipitation datasets of Mekis and Vincent (2011) (see Sect. 3.1) were carried out.*

- **Study area includes a discussion of data collection.**
  *We are unsure what the reviewer means by "a discussion of data collection" and we believe that we have discussed the overview of data availability in Canadian situation in the second paragraph of Section 2 [P7:L9-30] and we have also provided the data descriptions in Section 3 in the original manuscript. Also, we believe that it is better to separately describe the study area and data collection given the amount of datasets being analyzed which otherwise it will be too long for one section.*

- **Introduction should be presented on its own. "Precipitation measurements and their limitations" and "Objectives and Scope" should not be in the introduction.**
  *Thank you for your suggestion. We think that having the subheadings in the introduction helps the readers to better understand and to faster grasp the ideas of the paragraphs. Therefore, we decide to keep the subheadings in the revised manuscript.*

- **Most of section 3.2 can be removed and inserted as a summary table as it completely references the outcome of prior studies.**
  *The details in Section 3.1 and 3.2 will be greatly reduced in the revised manuscript and the changes are shown in the response to comment 2. We do have a summary table (Table 1) in the original manuscript to provide an overview of the datasets being compared.*

8. **Language: an edit should be conducted to check for grammar and sentence structure. Examples:**

**The results point on P28:15 contains 3 sets of parentheses in a single sentence.**

*We will delete one set of parentheses in the revised manuscript, which is shown as follows:*

*In northern Canada (above 60° N), the different products tended to moderately (ranging from -0.6 % to -40.3 %)  underestimate total precipitation, while reproducing the timing of daily precipitation rather well. It should be noted that this assessment was based on only a limited number of precipitation-gauges in the north.*

**The sentence on P7:L20 ends with "along the southern Canada".**

*We will change the sentence in the revised manuscript, which is shown as follows:*

*The Meteorological Service of Canada has implemented a network of 31 radars (radar coverage at full range of 256 km) along  southern Canada (see Fortin et al. (2015b) Fig. 1 for spatial distribution).*

**P8:L4 refers to the province of Alberta as Alberta province.**

*We will change "Alberta province" to "province of Alberta" in the revised manuscript, which is shown as follows:*

*Climate data collection is coordinated by the Federal government of Canada. Agriculture and Agri-Food Canada maintains a few stations nationally especially in  province of Alberta.*

9. **References: ample amount of references but this is appropriate given the amount of datasets being analysed. Though several references appear dated, for example the Radar Reflectivity and Surface Rainfall paper likely had several further advances on the topic since 1987.**

   *We agree that the Austin (1987) reference is a bit outdated and there are further advances in addressing the errors in rain-rate reflectivity by the radar. We will update and replace Austin (1987) reference in the revised manuscript by Villarini and Krajewski (2010), which is shown as follows:*

   Villarini, G., and Krajewski, W. F.: Review of the Different Sources of Uncertainty in Single Polarization Radar-Based Estimates of Rainfall, Surv Geophys, 31, 107-129, 10.1007/s10712-009-9079-x, 2010.

---

## Author Response (AR1)

**Responses to Editor final comments on Manuscript HESS-2016-511**

**Title:** Evaluation of various daily precipitation products for large-scale hydro-climatic applications over Canada

Authors: Jefferson Wong et al

Manuscript No: hess-2016-511

Dear Prof. Jan Seibert, thank you very much again for your comments and recommendations. We have addressed all of the comments and presented our responses below.

The review comments are in regular bold typeface, while all responses are in italics and indented paragraphs, with deleted materials being crossed out by drawing a line through them and revised sentences being coloured in red.

**Response to Editor**

Editor Decision: Publish subject to revisions (further review by Editor and Referees) (31 Dec 2016) by Prof. Jan Seibert

**Comments to the Author:**

The reviews list a number of important points and based on the responses in the discussion phase, I am confident that the authors will be able to address these. As also indicated by the reviews, the original submission suffered from language issues and from poor figures. The authors need to address these two issues also beyond the concrete suggestions throughout the manuscript. Here it is important that the senior authors (native speakers!) look carefully at the manuscript before resubmission!

In response to the Editor's comments, we have focused on proofreading the manuscript and made some additional modifications. We have gone through the entire manuscript and where applicable, have improved the language and flow (e.g., removed repetitive phrases, re-organized sub-sections). Specific areas where text has been modified include:

1) Sect. 3.1 [L227-270] now provides a better description of the precipitation-gauge station observations.

2) A new sub-section heading [Sect. 4.1; L483] has been added to include the description of the pre-processing of the gridded products and the precipitation-gauge station observations.

3) Two paragraphs have been re-arranged for better logical flow of the manuscript.

- a. Paragraph describing the precipitation measurements and their limitations in the Canadian context has been moved from Study Area Section [L203-224] to Introduction Section [L152-173].
- b. Paragraph describing the selection of the study period has been moved from [L519-526] to [L484-499].

**Response to Reviewer 1**

The authors evaluated various gridded precipitation datasets against long-term station data in order to assess accuracy of each datasets. The presence of multiple gridded precipitation datasets available to researchers these days, assessing the accuracy of these datasets (or more appropriately uncertainties in these available datasets) is a very important concern in hydrological research. From the point, this work can be a significant contribution to the hydrologic research for Canada. Despite its importance, I found a number of questions that need to be answered before this one is accepted for publication.

We are grateful to the reviewer for his/her review and comments and suggestions to improve our paper. We have now addressed all of the comments and presented our responses below.

**Specific comments:**

1. The authors compare gridded precipitation products against data at individual stations. A rain gauge data represents only a small area but the gridded data evaluated in this study, especially those based on model products, represent values at much larger area, essentially averages over individual grid boxes. How can we expect that a rain gauge data can represent an average value of hundreds of square kilometers? This must be thoroughly discussed to justify their methodology.

The reviewer's point is well-taken. We are aware of the challenges and issues with comparing point measurements and area-averaged estimates. However, in the absence of a sufficiently dense precipitation gauge network in Canada, our options for assessing the different gridded products would be limited. The only gridded product that is basically representing areal averages of precipitation (via interpolation) based on ground observations is ANUSPLIN. This product, however, may not be qualified as the "ground truth", as it has its own limitations which has already raised in the original manuscript (see Section 3.2.1). Therefore, we also included ANUSPLIN in the pool of gridded products to be evaluated.

Notwithstanding the issues, we found that using the selected gauge measurements would remain the best way for the evaluation of the multiple gridded products. First, this is because the set of gauges used has been adjusted (e.g. for undercatch) and are the most accurate source of information on precipitation in Canada (although with limited spatial coverage). Second, given that we compared all the gridded data products against this common set of point-based measurements, it may be safe to assume that the biases in differences between point and areal data is pretty much consistent for all the products. In other words, it would not work in favor of one product and against one other product. Parts of this discussion is already in the original manuscript (Section 6, L827-830). We have revised the manuscript to better reflect on this issue, which is shown as follows (coloured in red):

In addition, results from the above analysis should be interpreted with care because the precipitation-gauge station data are point measurements whereas the gridded precipitation products are areal averages, of which the accuracy and precision of the estimates could be very different given the non-linear responses of precipitation (Ebert et al., 2007). When comparing point measurements and areal-average estimates, fundamental challenges occur because of the sampling errors arising from different sampling schemes and errors related to gauge instrumentation (Bowman, 2005). It is therefore difficult to have perfect spatial matching between point measurements (gauge stations) and areal-averaged estimates (gridded products) (Sapiano and Arkin, 2009; Hong et al., 2007). However, in the absence of a sufficiently dense precipitation gauge network in Canada, the options for assessing different gridded products are limited. The only gridded product that essentially represents areal averages of precipitation (via interpolation) based on ground observations is ANUSPLIN. As aforementioned (see Sect. 3.2.1), this product has its own limitations and may not be qualified as the "ground truth". Therefore, ANUSPLIN is also included in the pool of gridded products to be evaluated. Notwithstanding the issues, the authors feel that using the selected gauge measurements is best for the evaluation of the multiple gridded products because the set of gauges used has been adjusted (e.g. for undercatch) and are the most accurate source of information on precipitation in Canada (although with limited spatial coverage). Also, given that all the gridded products are compared against this common set of point-based measurements, it is assumed that the biases in differences between point and areal data is consistent for all the products. However, the authors believe that given the current data situation, the preceding was the best methodology for evaluating the performance of different daily gridded precipitation products.

2. To prepare for evaluations, the authors first interpolated all gridded data into a common grid of 0.5deg resolutions, then they re-interpolated from the grid to the location of individual rain gauges. This data processing includes two spatial interpolations. Because every interpolation step introduces its own errors or uncertainties, the number of interpolation steps must be as small as possible. I wonder why they did not directly interpolate each data set to the rain gauge locations without going through the intermediate grid? This can simply data processing and can reduce interpolation-related uncertainties.

This point is also well-taken. But there are two things to consider. First of all, upscaling to a coarser grid size (e.g., from 10km to 50km) is mainly by averaging, and therefore, it would not introduce any significant errors into the upscaled data (unlike interpolation). Second, as the reviewer suggested, we could easily compare each gridded product against the point observations at their original resolutions. However, the main focus of this study is to inter-compare various gridded precipitation products using precipitation-gauge station data as a reference/benchmark but not to assess the individual accuracy of each product against the reference dataset. In other words, this study does not intend to assess different products for reproducing observed individual precipitation events generated by a given weather system but to examine the combined precipitation distribution over a period of time. Therefore, we opted to upscale them all to a common (coarser) grid size first. This way the inter-comparison would be more consistent, as the different products when brought to a common scale are expected to show more similar statistical properties. For example, by coarsening, you expect to reduce the temporal variability of data (manifested in variance) as well, while you may not want these differences due to having different spatial resolutions obscure your intercomparisons.

Moreover, the original spatial resolution of the climate forcing data does not always match with the spatial resolution of a large-scale hydrological model. Thus, under this circumstance, upscaling a fine resolution product is a necessary process for large-scale hydrological applications. The results of this study could therefore better reveal the errors incorporated in the rescaled precipitation products as the errors include the interpolation-related uncertainties. Also note that our methodology is consistent with the similar studies in the literature (e.g. Janowiak et al., 1998;Rauscher et al., 2010;Kimoto et al., 2005).

Accordingly, we have changed the title from "Evaluation" to "Inter-comparison" to better reflect our aim of the study. Also, we have summarized and inserted the above justification in the revised manuscript [L484-491], which is shown as follows:

Given that the main focus of this study was to inter-compare the various gridded precipitation products using precipitation-gauge station data as a reference/benchmark (and not to assess the individual accuracy of each product against this reference), it was decided to re-grid each product onto a common  $0.5^{\circ} \times 0.5^{\circ}$  resolution to match the lowest-resolution dataset. It was acknowledged that re-gridding can introduce uncertainties due to the extra interpolations, however, the authors believe that upscaling to a common resolution provided a direct and more consistent inter-comparison. Furthermore, this methodology was consistent with similar studies in the literature (e.g. Janowiak et al., 1998;Rauscher et al., 2010;Kimoto et al., 2005).

Lastly, motivated by this review comment, we conducted an inter-comparison test at the original scale (0.0833°) of ANUSPLIN against the upscale resolution (0.5°) in two ecozones (Ecozone 3 Southern Arctic and Ecozone 6 Boreal Shield where the numbers of precipitation-gauge stations are the least and largest, respectively), as shown in the following tables. The results show that the original and upscale resolutions produce performance measures of similar magnitude and the differences are not significantly large. Therefore, we believe that the interpolation-related uncertainties will be relatively smaller than the uncertainties arisen from other sources such as model structure, equifinality of parameters, and process representations.

| Region    |          | Resolution         |      |           |      |                         |      |           |      |
|-----------|----------|--------------------|------|-----------|------|-------------------------|------|-----------|------|
| (Ecozone) |          | Original (0.0833°) |      |           |      | $Upscale (0.5^{\circ})$ |      |           |      |
|           |          | 1979-2012          |      | 2002-2012 |      | 1979-2012               |      | 2002-2012 |      |
|           |          | PBias              | RMSE | PBias     | RMSE | PBias                   | RMSE | PBias     | RMSE |
| 3         | Southern | -24.16             | 1.83 | -36.88    | 2.37 | -27.89                  | 1.84 | -40.70    | 2.38 |
|           | Arctic   |                    |      |           |      |                         |      |           |      |
| 6         | Boreal   | -12.46             | 3.34 | -14.15    | 3.39 | -13.87                  | 3.35 | -15.67    | 3.40 |
|           | Shield   |                    |      |           |      |                         |      |           |      |

We believe including these results in the revised manuscript might be divergent from the core objective of this study. However, we would be open to suggestions by the reviewer.

3. Model products based on RCP scenarios includes the effects of hypothetical emissions pathways implemented in these simulations. How can these model data be compared against the reference data in the same wat as other assimilated and/or station-based gridded data? The authors evaluated these data sets for two periods, 1979-2012 and 2002-2012. The CMIP5 experiment that seem relevant to the model data used in this study was designed in such a way that the present-day period simulation based on the realistic GHG concentration for the period from mid-19th century to 2005. Future projections based on specific RCP scenarios starts from 2005 up to 2300 with the initial condition taken at the end of the present-day simulation period. Thus all model data after 2005 are affected by hypothetical emissions pathways. It's pretty unusual discussing "accuracy" of the data generated to project future on the basis of hypothetical GHG concentrations. If the authors are interested in evaluating the model-generated data, the comparison must end in 2005, the end of the present-day period for which the observed external forcing and GHG concentration are implemented. For such periods, there is no need to distinguish runs according to RCP scenarios because the hypothetical emissions pathways are not implemented. My suggestion is to drop model data from the evaluation, or include the model data and limited the evaluation period to 2005 instead of 2012.

We appreciate the value of the reviewer's comment on the climate model products and we agree that evaluating the model-generated products with different RCP scenarios is not appropriate. We decide to include the climate model products for evaluation and limit the evaluation period to 2005 instead of 2012. The reason of not dropping all the climate model products from the evaluation is we think it is still worthwhile to compare different climate model products to see which downscaling methods/which GCMs/RCMs provide better historical estimates so that potential users could use the results as a reference for their future climate change studies. Accordingly, the description of the evaluation period has been changed in the revised manuscript [L492-496], which is shown as follows:

Two common time spans were selected since CaPA covered a shorter time frame when compared to the rest of the products: (1) long-term comparison from January 1979 to December 2012 with the exclusion of CaPA (from January 1979 to December 2005 for PCIC and NA-CORDEX as the historical period of the datasets ends in 2005); and (2) short-term comparison from January 2002 to December 2012 when CaPA are available.

Also, the percentage of reliability has been re-calculated for the climate model products in the revised manuscript and the results have been revised in the Results Section (Section 5.1), which is shown as follows:

Regarding the PCIC ensembles, the different GCMs provided a range of reliabilities for the individual seasons. GFDL-ESM2G performed the best in spring (58.6 %) while CanESM2 in autumn (43.8 %). MPI-ESM-LR generally gave more reliable estimates in summer and winter (64.5 % and 38.3 %).MPI-ESM-LR performed the best in summer (70.2 %) while CanESM2 in autumn (45.5 %). GFDL-ESM2G generally gave more reliable estimates in spring and winter (57.4 % and 41.7 %). The performance of HadGEM2 ES RCP 8.5 with BCCAQ statistical downscaling method was significantly poorer than the rest of the GCM ensembles, especially in summer (13.1 %). Overall, the performance of MPI-ESM-LR (49.1 %52.0 %) was the best among the GCMs, followed by GFDL-ESM2G (47.0 %50.1 %), CanESM2 (42.2 %47.8 %), and HadGEM2 (36.7 %36.2 %). In terms of statistical downscaling methods, the BCCAQ method was on average slightly better than BCSD (47.5% versus 45.4 %49.5 % versus 44.0 %) with the former having a greater similarity in spring and summer as opposed to autumn and winter. These small differences therefore suggest that both methods are similar. With respect to the NA-CORDEX ensembles, the CRCM5 RCM gave the most reliable estimates in summer and autumn regardless of the GCM used. CanRCM4 had the best reliability in spring (46.9 %49.4 %) whereas RegCM4 had the poorest reliability in spring and summer (22.1 % and 36.6 %24.4 % and 34.0 %). In addition, the CanESM2 driven CanRCM4 with RCP 4.5 and RCP 8.5

[revised manuscript text omitted]

*The Discussion Section (Section 6) related to the climate model products has also been revised to reflect the change [L797-812], which is shown as follows:*

By matching the statistical property properties of the adjusted gauge measurements at monthly time scale, one could establish the confidence in using the climate model-simulated products for long-term hydro-climatic studies. Comparing the overall reliability of the PCIC and NA-CORDEX datasets, it was found that for the individual seasons the PCIC ensembles (from spring to winter: 52.2 %, 56.0 %, 41.9 %, and 32.4 % spring, summer, and winter: 54.0 %, 64.7 %, and 35.7 %) outperformed the NA-CORDEX ensembles (34.5 %, 41.4 %, 38.3 %, and 31.7 %39.1 %, 45.0 %, and 31.3 %) under RCP 8.5 scenario. This result was the same under RCP 4.5 scenario except in autumn when the NA-CORDEX ensembles (46.2 %45.5 %) provided slightly higher reliability than the PCIC ensembles (42.5 %45.2 %). The better reliability of the PCIC datasets could be due to the use of ANUSPLIN to train the GCMs and thus, the statistical properties of the downscaled outputs are guided by those of the ANUSPLIN. Similarly, for ecozones where more than 10 precipitation-gauge stations could be found (Regions 6 to 9, 13 and 14), the PCIC ensembles (reliability ranging from <del>36.4 %</del> to 68.1 %35.7 % to 64.4 %) also outperformed the NA-CORDEX ensembles (from 16.8 % to 49.9 %17.2 % to 61.6 %). This would suggest that the PCIC ensembles may be the preferred choice for long-term climate change impact assessment over Canada, although further research is required.

Please note that the re-calculation does not affect the overall conclusion we made in the original manuscript.

4. The authors provide lengthy descriptions on the details of the data sets used in this study. Much of these discussions are unnecessary because there were developed by other research groups and relevant publications on the details of these data sets are already available. Sections 3.1 and 3.2 can be reduced by referencing suitable publications.

The details in Sections 3.1 and 3.2 have been greatly reduced in the revised manuscript. In short, the spatial and temporal resolutions of each product, their compositions, and examples of their applications are remained and other details have been deleted. The following shows the revised Sections 3.1 and 3.2, with deleted materials being crossed out by drawing a line through them (and revised sentences being coloured in red):

**3.1 Precipitation-gauge station data-observations**

In Canada, Climate climate data collection is coordinated by the Federal government, which is of Canada. Agriculture and Agri-Food Canada maintains a few stations nationally especially in Alberta province. Also, most hydro-power companies collect their own data. However, their data are not made available by to the public but are sent to Environment and Climate Change Canada for archiving prior to release. In other words, the National Climate Data Archive of Environment and Climate Change Canada (NCDA). These data provide the basis for all the available quality controlled climate data observations. Based on the National Climate Data Archive of Environment *Canada, there There are a total of 1499 precipitation-gauge stations (as in of 2012)* across Canada. However, due to the given the frequent addition and subtraction of climate stations, these numbers have greatly varied through time with peak reporting in the 1970s followed by a general decline to the present over the past few decades, the number of stations with available precipitation data for specified time intervals varies greatly. For instance, the numbers of precipitation gauge stations that were active in any given years over the period of 1961 to 2003 ranged from 2000 to 3000 (see Hutchinson et al. (2009) Figs 1 and 2 for details). The issue with these data is they are Furthermore, the existing precipitation observations are often subject to various errors, with gauge undercatch being of significant concern among which the errors due undercatch are quite significant in Canada (Mekis and Hogg, 1999). In order to To account for various measurement issues, Mekis and Hogg (1999) first produced the Adjusted and Homogenized Canadian Climate Data (AHCCD) including adjusted daily rainfall and snowfall values and Mekis and Vincent (2011) then updated the data for a subset of 464 stations over Canada. provided adjusted daily rainfall and snowfall data for 464 stations over Canada that were based on the Adjusted Precipitation for Canada dataset (Mekis and Hogg, 1999). The data extend back to 1895 for a few long-term stations and run through 2014. For these data, daily rainfall gauge and snowfall ruler data were extracted from the National Climate Data Archive of Environment Canada and adjustments of rain and snow were done separately. Regarding each rain gauge type, corrections for wind undercatch, evaporation and wetting losses were performed based on field experiments at various locations (Devine and Mekis, 2008). For snowfall, a density correction based on coincident ruler and Nipher gauge observations was applied to all snow measurements (Mekis and Brown, 2010). Adjustments were also implemented to account for trace precipitations and accumulated amounts from multiple days were distributed over the affected days to minimize the impact on extreme values and preserve the monthly totals. Observations from nearby stations were

sometimes combined to create longer time series and adjustments were done either based on overlapping observations or standardized ratios between test sites and their neighbours (Vincent and Mekis, 2009). As a result of adjustments, total rainfall amounts were concluded to be on the order of 5 to 10 % higher in southern Canada and more than 20 % in the Canadian Arctic when compared to than the original observations. The effect of the adjustments on Adjustments to snowfall were even larger and more variable varied throughout the country. Despite the lack of a measure of associated uncertainty, this adjusted precipitation gauge station dataset has been recognized and widely used for different These adjusted values are considered as better estimates of actual precipitation and therefore have been used in numerous analyses (e.g. Nalley et al., 2012; Shook and Pomeroy, 2012; Wan et al., 2013). Therefore, this dataset was used in this study as the reference to represent the best available precipitation measurement and Given the lack of an adjusted daily gridded precipitation data for Canada, the AHCCD station precipitation is considered to be the best available data for Canada and thus is used as the benchmark for all gridded precipitation product comparisons.

**3.2 Gridded precipitation products**

Seven precipitation datasets were assessed. Table 1 provides a concise summary of these datasets, including their full names, and original spatial and temporal resolutions for the versions used. These particular datasets were chosen for assessment based on the following criteria: (1) a complete coverage of Canada; (2) minimum of daily temporal and  $0.5^{\circ}$  (~50 km) spatial resolutions; (3) sufficient lengths of data (>30 years) for long-term study and cover including recent years up to 2012; and (4) representation of representing a range of sources/methodologies (e.g. station based, remote sensing, model, blended products). Table 1 summarizes these datasets, including their full names and original spatial and temporal resolutions for the versions used. Note that other commonly used datasets including the monthly Canadian Gridded temperature and precipitation (CANGRD) dataset (Zhang et al., 2000), and the coarser resolution Japan Meteorological Agency 55-year Reanalysis (JRA-55) (Onogi et al., 2007;Kobayashi et al., 2015), and the Modern-Era Retrospective Analysis for Research and Applications (MERRA) (Rienecker et al., 2011) products were excluded as they do not meet criteria # 2 (2) above.

**3.2.1 Station-based product – ANUSPLIN**

With the application of the Australian National University Spline (ANUSPLIN) model (Hutchinson, 1995;Hutchinson, 2004), Hutchinson et al. (2009) used the Australian National University Spline (ANUSPLIN) model to developed develop a climate dataset of daily precipitation and daily minimum and maximum air temperature over Canada at a spatial resolution of 300 arc-seconds of latitude and longitude (0.0833° or ~10 km) for

the period of 1961 to 2003, using observed stations. All available NCDA stations (that ranged from 2000 to 3000 in for any given years over the during this period) were used as an input to the gridding procedure. recorded in the National Canadian Climate Data Archives of Environment Canada. However, to To retain a better maximum spatial coverage, the smaller number of stations in AHCCD were not incorporated (i.e. only unadjusted archive values were used). no adjustments were done on the archive station data before the generation of the product. The dataset was generated to model the complex spatial patterns by using Interpolation procedures included incorporation of tri-variate thin-plate smoothing splines method that incorporated using spatially continuous functions of latitude, longitude, and elevation. Hopkinson et al. (2011) subsequently extended this original dataset to include the period of 1950 to 2011. This ANUSPLIN product for Canada (hereafter the ANUSPLIN) has first been quality controlled with various flags indicating trace values, accumulated values over multiple days, and missing and estimated values. The accuracy of the product was then assessed by withholding from the analyses 50 stations broadly representing the southern half of Canada and by examining the error statistics for the withheld stations. The ANUSPLIN dataset-The Canadian ANUSPLIN has now further been updated to 2013 and has recently been used as the basis of 'observed' data for evaluating different climate datasets (e.g. Eum et al., 2012) and for assessing the effects of different climate products in hydrological applications (e.g. Eum et al., 2014; Bonsal et al., 2013;Shrestha et al., 2012a).

**3.2.2 Station-based model-derived multiple-source product – CaPA**

Initiated in In November 2003 through collaborations within the Meteorological Service of Canada, the Canadian Precipitation Analysis (CaPA) was developed to produce a dataset of 6-hourly precipitation accumulation over North America in realtime at a spatial resolution of 15 km (from 2002 onwards) (Mahfouf et al., 2007). The dataset was Data were generated based on-using an optimum interpolation technique (Daley, 1993), which required a background field and a specification of error statistics between the observations and the *a* background field (e.g. Bhargava and Danard, 1994; Garand and Grassotti, 1995). For Canada, the short-term precipitation forecasts from the Canadian Meteorological Centre (CMC)'s regional model, the Global Environmental Multiscale (GEM) model (Cote et al., 1998a;1998b), were used as the background field with the rain-gauge measurements from the observational network NCDA as the observations to generate an analysis error at every grid point. The analysis was created by simple kriging to interpolate the differences between the transformed data of GEM and stations, which was then re-transformed and applied back to GEM. The quality of rain gauge stations was controlled by cross-checking with the neighbouring stations and by comparing with the radar derived precipitation. The accuracy of the product was assessed by generating an analysis error that represented

the amount of additional information gained from the multiple observations with regard to the background field. CaPA has become operational at the CMC in April 2011, with updates to in the statistical interpolation method (Lespinas et al., 2015), and increase of spatial resolution to 10 km. and the The assimilation of Quantitative Precipitation Estimates from the Canadian Weather Radar Network is also used as an additional source of observations (Fortin et al., 2015b). With its continuous improvement and different configurations, CaPA has been employed in Canada for various environmental prediction applications (e.g. Eum et al., 2014;Fortin et al., 2015a;Pietroniro et al., 2007;Carrera et al., 2015). However, the study period of these applications only extended back to started in 2002.

3.2.3 Reanalysis-based multiple-source products – Princeton, WFDEI, and NARR

**Princeton**

The Terrestrial Hydrology Research Group at the Princeton University initially developed a dataset of 3-hourly near-surface meteorology with global coverage at  $\frac{1}{4}$  $1.0^{\circ}$  spatial resolution (~120 km) from 1948 to 2000 for driving land surface models and other terrestrial systems (Sheffield et al., 2006). The global dataset at the Princeton University This dataset (called hereafter the "Princeton") was constructed based on the National Centers for Environmental Prediction-National Center for Atmospheric Research (NCEP-NCAR) reanalysis (2.0° and 6-hourly) (Kalnay et al., 1996;Kistler et al., 2001), combining with a suite of global observation-based data including the Climatic Research Unit (CRU) monthly climate variables (New et al., 2000, 1999), the Global Precipitation Climatology Project (GPCP) daily precipitation (Huffman et al., 2001), the Tropical Rainfall Measuring Mission (TRMM) 3-hourly precipitation (Huffman et al., 2002), and the NASA Langley Research Center monthly surface radiation budget (Gupta et al., 1999). Regarding precipitation, the dataset has undergone several stages in terms of spatial downscaling with the use of GPCP data, temporal downscaling based on sampling from TRMM data, and the sophistication of the correction methods (a correction to the wet-day statistics (Sheffield et al., 2004), and monthly bias corrections to match those of the CRU data (Adam and Lettenmaier, 2003)). The Princeton dataset has been evaluated against the Second Global Soil Wetness Project (GSWP-2) product (Zhao and Dirmeyer, 2003). With the inclusion of new-additional temperature and precipitation data (e.g. Willmott et al., 2001), Princeton has been updated and is currently available with two versions: 1) 1948 to 2008 at 1.0°, (plus 0.5°, and 0.25°), at 3-hourly, (plus daily, and monthly) resolution globally for 1948 to 2008 time steps and 2). Experimental updates including a 1901-2012 experimental version at 1.0°, (plus 0.5°), at 3-hourly, (plus daily, and monthly) resolution are also available time steps (used in this study). Studies employing Princeton to study examine different hydrological aspects have been carried out over different parts of Canada (e.g. Kang et al., 2014;Su et al., 2013;Wang et al.,

2013; Wang et al., 2014). For instance, Kang et al. (2014) examined the changing contribution of snow to runoff generation in the Fraser River Basin while Su et al. (2013) investigated the relationships between spring snow and warm-season precipitation in central Canada. In addition, Wang et al. (2013) and Wang et al. (2014) used this dataset to characterize the spatial and seasonal variations of the surface water budget at Canada national scale.

**WFDEI**

To simulate the terrestrial water cycle using different land surface models and general hydrological models, the European Union Water and Global Change (WATCH) Forcing Data (WFD) were created to provide datasets of sub-daily (3-hourly or and 6hourly) and daily meteorological data with global coverage at  $a 0.5^{\circ}$  spatial resolution (~50 km) from 1901 to 2001 (Weedon et al., 2011). Similar to the composition of the Princeton dataset, the WFD were derived from the 40-year European Centre for *Medium-Range Weather Forecasts (ECMWF) Re-Analysis (ERA-40) (1.0° and 3-hourly)* (Uppala et al., 2005) and combined with the CRU monthly variables and the Global Precipitation Climatology Centre (GPCC) monthly data (Rudolf and Schneider, 2005; Schneider et al., 2008; Fuchs, 2009). The generation of the WFD for 1958 to 2001, which was based on the ERA 40, followed the procedures developed by Ngo Duc et al. (2005) and Sheffield et al. (2006) whereas the dataset for 1901 to 1957 was generated by using the reordered ERA-40 a year at a time. With respect to precipitation, the creation of the data (Weedon et al., 2010) involved spatially downscaling using the CRU data, sequential elevation correction, wet day correction, monthly precipitation bias correction to match the GPCC data, and adjustment for gauge undercatch (Adam and Lettenmaier, 2003), however no corrections were made for orography effect (Adam et al., 2006). The same monthly bias corrections were also done using the CRU precipitation totals, resulting in two sets of precipitation data. The WFD were assessed by the FLUXNET data for selected years at seven sites (Araujo et al., 2002; Persson et al., 2000; Suni et al., 2003; Meyers and Hollinger, 2004; Grunwald and Bernhofer, 2007; Urbanski et al., 2007; Gockede et al., 2008). The WATCH Forcing Data methodology applied to ERA-Interim (WFDEI) dataset has further been generated developed covering the period of 1979 to 2012 (Weedon et al., 2014). The WFDEI used the same methodology as the WFD, but was based on the ERA-Interim (Dee et al., 2011) with higher spatial resolution  $(0.7^{\circ})$ , better data assimilation technique, updated monthly observation based data, more extensive incorporation of observations, and correction of the most extreme cases of inappropriate precipitation phase. As for the WFD, the WFDEI had two sets of rainfall and snowfall data generated by using either CRU or GPCC precipitation totals. Both sets of data were used in this study (hereafter the known as WFDEI [CRU] and WFDEI [GPCC], respectively). To date, specific studies using the WFDEI related to Canada has been limited to the studies of

permafrost in the Arctic regions (e.g. Chadburn et al., 2015;Park et al., 2015;Park et al., 2016) but the WFDEI could be a potential source in other environmental applications in Canada.

**NARR**

Concerning the With the aim of evaluating spatial and temporal water availability in the atmosphere, the North American Regional Reanalysis (NARR) was developed to provide datasets of 3-hourly meteorological data for the North America domain at a spatial resolution of 32 km (~0.3°) covering the period of 1979 to 2003 as the retrospective system and is being continued in near real-time (currently up to 2015) as the Regional Climate Data Assimilation System (R-CDAS) (Mesinger et al., 2006). The components in generating NARR included the NCEP-DOE reanalysis (Kanamitsu et al., 2002), the NCEP regional Eta Model (Mesinger et al., 1988;Black, 1988) and its Data Assimilation System, a recent version of the Noah land-surface model (Mitchell et al., 2004; Ek et al., 2003), and the use of numerous additional data sources (see Mesinger et al., 2006 Table 2). The use of NCEP-DOE reanalysis was a major improvement upon the earlier NCEP-NCAR reanalysis in both resolution and accuracy to provide lateral boundary conditions. Regarding precipitation assimilation scheme, the NARR adjusted the accumulated convective and grid scale precipitation, assimilated the precipitation observations as latent heating profiles based on the differences between the modelled and observed precipitation (Lin et al., 1999), and disaggregated into hourly resolution using different sources over lands and oceans. For the period from 1979 to 2003 when NARR was run as the retrospective system, precipitation analyses over the continental United States (CONUS), Mexico, and Canada were derived solely from a gridded analysis of 24 hour rain gauge measurements. For the period from 2004 onwards, NARR was generated in near real time by the R CDAS, which was identical to the retrospective NARR except for changes in input sources and their processing because of the real time production constraints. One of the major differences was the use of radardominated precipitation analyses derived from the National Land Data Assimilation System (NLDAS) (Mitchell et al., 2004) over CONUS to disaggregate the 24-hour raingauge analysis to hourly precipitation whereas no assimilation was done over Canada due to the paucity of rain gauge observations. On the basis of For hydrological modelling in Canada, Choi et al. (2009) found that NARR provided reliable climate inputs for northern Manitoba while Woo and Thorne (2006) concluded that NARR had a cold bias resulting in later snowmelt peaks in subarctic Canada. In addition, Eum et al. (2012) identified a structural break point in the NARR dataset beginning in January 2004 over the Athabasca River basin due to the assimilation of station observations over Canada being discontinued in 2003.

3.2.4 GCM statistically downscaled products – PCIC

The Pacific Climate Impacts Consortium (PCIC), which is a regional climate service centre at the University of Victoria, British Columbia, Canada, has offered datasets of statistically downscaled daily precipitation and daily minimum and maximum air temperature under three different Representative Concentration Pathways (RCPs) scenarios (RCP 2.6, RCP-4.5, and RCP-8.5) (Meinshausen et al., 2011) over Canada at a spatial resolution of 300 arc-seconds (0.833° or ~10 km) for the historical and projected period of 1950 to 2100 (Pacific Climate Impacts Consortium; University of Victoria, Jan 2014). These downscaled datasets were a composite of 12 GCM projections from the Coupled Model Inter-comparison Project Phase 5 (CMIP5) (Taylor et al., 2012) and the ANUSPLIN dataset. The historical 1950 to 2005 period of the ANUSPLIN was used for bias-correction and downscaling of the GCMs. to drive the GCMs and the statistical properties and spatial patterns of the downscaled outputs tended to resemble those of the ANUSPLIN. However, the timing of natural climate variability (e.g. El Niño Southern Oscillation) in the observational record were not considered since GCMs were solved as a 'boundary value problem'. Two different downscaling methods were used to downscale to a finer resolution (Werner and Cannon, 2016). The first one was These included Bias Correction Spatial Disaggregation (BCSD) (Wood et al., 2004) following Maurer and Hidalgo (2008) and the second was Bias Correction Constructed Analogues (BCCA) with Quantile mapping reordering (BCCAQ) which was a post-processed version of BCCA (Maurer et al., 2010). In general, the most important distinction between the two methods was BCCAO obtained spatial information from a linear combination of historical analogues for daily values and retained the daily sequencing of weather events from the coarse resolution, while BCSD only used monthly averages to reconstruct daily patterns by randomly resampling a historic month and scaling its daily values to match the monthly projected values. The ensemble of the PCIC dataset has currently been used in studying the hydrological impacts of climate change on river basins mainly in British Columbia (e.g. Shrestha et al., 2011; Shrestha et al., 2012b; Schnorbus et al., 2014) and Alberta (e.g. Kienzle et al., 2012; Forbes et al., 2011) in Canada. In this study, only four GCMs with two respective statistically downscaling methods under RCP 4.5 and 8.5 were chosen for comparison (see Table 2 for details). The choice of selecting the four GCMs under RCP 4.5 and 8.5 only in the PCIC dataset was to match those GCMs available in the NA-CORDEX dataset (see next section for details).

3.2.5 GCM-driven RCM dynamically downscaled products – NA-CORDEX

Sponsored by the World Climate Research Programme (WCRP), the COordinated Regional climate Downscaling EXperiment (CORDEX) over North America domain (NA-CORDEX) was launched to provide dynamically downscaled datasets of 3-hourly or daily meteorological data over most of North America (below 80° N) at two spatial resolutions of 0.22° and 0.44° (or 25 and 50 km) under two different RCPs (RCP 4.5;

and RCP-8.5) for the historical (1950 - 2005) and projected future (2006 - 2100)period of 1950 to 2100 (Giorgi et al., 2009). Within the NA CORDEX framework, a matrix of six GCMs from the CMIP5 driving six different RCMs was selected to compare the performance of RCMs and characterize the uncertainties underlying regional climate change projections and thus provided climate scenarios for further impact and adaption studies. On top of the knowledge and experience gained from Drawing from the strengths of the North American Regional Climate Change Assessment Program (NARCCAP) (Mearns et al., 2012), a matrix of six GCMs from the CMIP5 driving six different RCMs was selected to compare and characterize the uncertainties of RCMs and thus provided climate scenarios for further impact and adaption studies. the selection of GCM RCM matrix of simulations, with higher spatial resolution and greater sampling of uncertainty, was based on model climate sensitivity and quality of boundary conditions. In addition, to determine the large variations in future climate due to internal variability of the GCMs on downscaled outputs, samples among multiple realizations of GCM simulations were used to drive the RCMs. The performance of participating RCMs in reproducing historical and projected climate was then assessed by comparing the ERA Interim driven RCM simulations. Current studies using NA-CORDEX datasets were mainly focused on evaluating the model performance of different GCM-driven RCM simulations over North America (e.g. Lucas-Picher et al., 2013; Martynov et al., 2013; Separovic et al., 2013) but the NA CORDEX dataset could also be a potential source in hydro-climatic studies in Canada. In this study, only two GCMs with and three RCMs were chosen for comparison due to the availability of the NA-CORDEX dataset (see Table 3 for details).

**5. All figures are too busy to read. Need to make them bigger.**

We believe that Figures 1, 5 to 8 are clear enough to show the messages and therefore we have only enlarged the figures as much as possible in the revised manuscript. In response to comment 3, we decide to include the climate model products for evaluation and limit the evaluation period to 2005 instead of 2012. Accordingly, Figures 2, 3, and 4 in the original manuscript have been reproduced to reflect the change. In short, the evaluation for the climate model products from the period of 1979 to 2005 will be shown separately from that of station-based and reanalysis-based products. Thus, Figures 3 and 4 will only show the distributions of p-value of the K-S test for the station-based and reanalysis-based products and a new Figure 5 will be created to show the distributions of p-value of the K-S test for climate model products in the revised manuscript. The numbering of Figures 5 to 8 will also be changed accordingly. Note that all the figures in the supplementary materials have also been subject to the same changes as aforementioned but will not be shown here. The revised figures are shown as follows:

---

## Referee Report (RR1)

**Evaluation of various daily precipitation products for large-scale hydro-climatic applications over Canada**

**The main areas for revision in the updated manuscript are run on sentences/comma errors, overwhelming supplemental information and results presentation. The methodology has been presented clearly and overall the section is clear to follow. Furthermore, the discussion is also presented well and highlights the important information derived from the results.**

**Points for revision:**

- The sentence at the very beginning of the paper is run on. The reader would have an easier time processing the information if it was divided into two parts.

*"This study inter-compares several gridded precipitation products and quantifies the spatial and temporal variability of the errors (relative to station observations) over 15 terrestrial ecozones in Canada for different seasons over the period 1979 to 2012 at a 0.5° and daily spatiotemporal resolution"*

- A comma is missing after *precipitation*.

*"The availability of accurate data, especially precipitation is essential for understanding the climate system and hydrological processes since it is a vital element of the water and energy cycles and a key forcing variable for driving hydrological models"*

- The comma is not needed after *part*

*"It is interesting to note that for the most part, there is a higher percentage of reliability in short-term period compared to long-term period."*

- The entire paper would benefit from a thorough review to correct these types grammatical errors as sentence structure can drastically alter the meaning of a statement.

- The section on precipitation measurements and their limitations is a very lengthy amount of background information that doesn't necessarily contribute to the goal of the paper which is an inter-comparison of precipitation products.

- Another example where information can be removed is where the requirements to choose the 7 products are stated clearly, but then datasets which do not meet the requirements are mentioned. It is obvious for a reader to understand that if something did not meet the requirements it would not be included.

*"Note that other commonly used datasets including the monthly Canadian Gridded temperature and precipitation (CANGRD) (Zhang et al., 2000), the coarser resolution Japan Meteorological Agency 55-year Reanalysis (JRA-55) (Onogi et al., 2007;Kobayashi et al., 2015), and the Modern-Era Retrospective Analysis for Research and Applications (MERRA) (Rienecker et al., 2011) products were excluded as they do not meet criteria (2) above."*

- In the following paragraph only the information pertaining to this study and the dataset used needs to be included. It can be reduced to one line.

*"1) 1948 to 2008 at 1.0°, 0.5°, and 0.25° at 3-hourly, daily, and monthly time steps and 2) 1901-2012 experimental version at 1.0° and 0.5° at 3-hourly, daily, and monthly time steps (used in this study). Studies employing Princeton to examine different hydrological aspects have been carried out over different parts of Canada. For instance, Kang et al. (2014) examined the changing contribution of snow to runoff generation in the Fraser River Basin while Su et al. (2013) investigated the relationships between spring snow and warm-season precipitation in central Canada. In addition, Wang et al. (2013) and Wang et al. (2014) used this dataset to characterize the spatial and seasonal variations of the surface water budget at Canada national scale"*

- The figures are crowded and do not present the information in a manner which is useful to the reader (even on a presentation screen the key information was impossible to decipher). The results section is also lengthy and important values are lost amongst the words. As each of the performance measures results in a value it would make better sense to present the results in a tabular format. This would alleviate the issue of information being lost and help the reader gain a clear picture of the performance as they could on their own compare values.

---

## Author Response (AR2)

**Responses to Editor final comments on Manuscript HESS-2016-511**

**Title:** Inter-comparison of daily precipitation products for large-scale hydro-climatic applications over Canada

Authors: Jefferson Wong et al

Manuscript No: hess-2016-511

Dear Prof. Jan Seibert, thank you again for your comments and recommendations. We have addressed all of the comments and presented our responses below.

The review comments are in regular bold typeface, while all responses are in italics and indented paragraphs, with deleted materials being crossed out by drawing a line through them and revised sentences being coloured in red.

**Response to Editor**

Editor Decision: Publish subject to minor revisions (further review by Editor) (21 Feb 2017) by Prof. Jan Seibert

**Comments to the Author:**

Thanks for your efforts with revising the manuscript. Reviewer #2 provides some useful comments, which will help you to further improve the manuscript. A critical issue are the figures, which still are not really satisfactory, if I may say. Especially for figures 3-5 a better design is needed. The small plots make it really difficult for your reader to get the information you want to show.

In response to the Editor's comments, we have excluded the regions where the number of stations in ecozone are less than 10 in the figures for better information delivery. Accordingly, Figures 3, 4, and 5 only showed regions having more than or equal to 10 stations (6 to 9 and 13, 14) in box-whisker plots for illustration. Note that Figures S2-S4 in the supplementary materials have also been subject to the same changes as aforementioned but will not be shown here. The revised figures are shown as follows:

Figure 1. Distributions of p-value of the K-S test in four seasons for the period of 1979 to 2012 (long-term comparison without CaPA). Note that the numbers of precipitation-gauge stations in each ecozone are different (see Table 4). The p-values of Regions 6 to 9, and 13 to 14 (R6-R9, and R13-R14), which have more than or equal to 10 stations, were only shown for illustration in box-whisker plots with bottom, band (black thick line) and top of the box indicating the 25th, 50th (median), and 75th percentiles, respectively.